# Controlling Statistical, Discretization, and Truncation Errors in Learning Fourier Linear Operators

**Unique Subedi**                                                                                                    *subedi@umich.edu*

**Ambuj Tewari**                                                                                                     *tewaria@umich.edu*

*Department of Statistics, University of Michigan*

**Reviewed on OpenReview:** `https://openreview.net/forum?id=A2sHNGcjLO`

## Abstract

We study learning-theoretic foundations of operator learning, using the linear layer of the Fourier Neural Operator architecture as a model problem. First, we identify three main errors that occur during the learning process: statistical error due to finite sample size, truncation error from finite rank approximation of the operator, and discretization error from handling functional data on a finite grid of domain points. Finally, we analyze a Discrete Fourier Transform (DFT) based least squares estimator, establishing both upper and lower bounds on the aforementioned errors.

## 1 Introduction

In operator learning, the goal is to use statistical methods to estimate an unknown operator between function spaces. A primary application of operator learning is the development of fast data-driven methods to approximate the solution operator of partial differential equations (PDEs) (Li et al., 2021; Kovachki et al., 2023). For example, consider the heat equation

$$\frac{\partial u}{\partial t} = \tau \, \nabla^2 u,$$

where $u : [0,1]^d \to \mathbb{R}$ vanishes on the boundary. The solution operator for this equation is a linear operator $\exp(\tau t \nabla^2) := \sum_{k=0}^{\infty} (\tau t \, \nabla^2)^k / k!$. Fixing some time point (say $t = 1$), our objective is to learn the solution operator $\mathcal{L} := \exp(\tau \nabla^2)$.

Given the training data $(v_1, w_1), \ldots, (v_n, w_n)$ where $w_i = \mathcal{L} v_i$, operator learning entails using statistical methods to estimate the solution operator $\widehat{\mathcal{L}}_n$. Then, given a new input $v$, one can get the approximate solution $\widehat{w} = \widehat{\mathcal{L}}_n v$. The goal is to develop the estimation rule such that $\widehat{w}$ is close to the actual solution $w = \mathcal{L} v$ under some appropriate metric.

Traditionally, given an input function $v$, one would use numerical methods such as finite differences to get a numerical solution. The solver starts from scratch for every new function $v$ of interest and can be computationally slow and expensive. This can be limiting in some applications such as engineering design where the solution needs to be evaluated for many different instances of the input functions. To solve this problem, operator learning aims to learn surrogate models that significantly increase speed for solution evaluation compared to traditional solvers while sacrificing a small degree of accuracy.

In this work, rather than focusing on specific PDEs, we adopt a broader perspective and study the learning-theoretic foundations of operator learning. For this task, we use the linear layer of the influential Fourier Neural Operator (FNO) architecture proposed by Li et al. (2021) as our model problem. While our results offer some practical and theoretical insights into the FNO, it is important to emphasize that our primary

objective is neither to advance the practical implementation of operator learning nor to develop deeper insights into the FNO architecture itself. Instead, our objective is to rigorously understand the statistical learning aspects of the operator learning paradigm. Our primary goal is to understand how operator learning differs from traditional machine learning settings and to identify the new techniques required to build a rigorous learning-theoretic foundation for this emerging area.

To this end, we start by identifying the distinct types of errors that are unique to operator learning. In addition to the standard statistical error arising from a finite sample size, operator learning introduces a discretization error due to the functional data being available only on a finite grid of domain points. Furthermore, ignoring high-frequency Fourier modes lead to a truncation error. Lastly, we introduce a Discrete Fourier Transform (DFT)-based estimator for our model problem and demonstrate how these errors can be systematically quantified for this estimator.

## 1.1 Neural Operators

To formally define our problem setting, we need to introduce neural operators from (Kovachki et al., 2023). Let $\mathcal{V}$ be a vector space of functions from a bounded subset $\mathcal{X} \subseteq \mathbb{R}^d$ to $\mathbb{R}^p$, and $\mathcal{W}$ to be a vector space of functions from $\mathcal{Y} \subseteq \mathbb{R}^d$ to $\mathbb{R}^q$. Given a function $v \in \mathcal{V}$, a single layer of neural operator $\mathtt{N}_t : \mathcal{V} \to \mathcal{W}$ is a mapping such that

$$(\mathtt{N}_t v)(y) = \sigma\Big( \left(\mathcal{K}_{\theta_t} v\right)(y) + b_t(y) \Big) \qquad \forall y \in \mathcal{Y},$$

where $\left(\mathcal{K}_{\theta_t} v\right)(y) = \int_{\mathcal{X}} k_{\theta_t}(y, x)\, v(x)\, dx$. The function $b_t : \mathcal{Y} \to \mathbb{R}^q$ is a bias function in $\mathcal{W}$, the function $\sigma : \mathbb{R}^q \to \mathbb{R}^q$ is a point-wise non-linear activation, and the transformation $v \mapsto \mathcal{K}_{\theta_t} v$ is an integral kernel transform of $v$ using some kernel $k_{\theta_t} : \mathcal{Y} \times \mathcal{X} \to \mathbb{R}^{q \times p}$. These layers are then composed to get a neural operator architecture.

Parametrizing $\mathcal{K}_{\theta_t}$ in terms of $k_{\theta_t}$ can be impractical due to the computational cost of calculating the integral in for each layer. Thus, a significant area of research in neural operators focuses on developing innovative parametrizations of $\mathcal{K}_{\theta_t}$ that facilitate more efficient computation. One such parametrization gives rise to a well-known architecture called the Fourier Neural Operator.

## 1.2 Fourier Neural Operator (FNO)

In this section, we present a brief, non-rigorous overview of the FNO. A more formal treatment, along with new insights into its parametrization, is provided in Appendix B.

We consider the setup from the work of Li et al. (2021). Let $\mathcal{X} = \mathcal{Y} = \mathbb{T}^d \simeq [0,1]^d$ be a $d$-dimensional periodic torus. Assume the kernel $k_\theta$ is translation invariant–that is, $k_\theta(y, x) = k_\theta(y - x)$. This implies that $\mathcal{K}_\theta$ is a convolution operator. Then, the Convolution Theorem implies that

$$\mathcal{K}_\theta v = \mathcal{F}^{-1}\Big( \mathcal{F}(k_\theta)\, \mathcal{F}(v) \Big),$$

where $\mathcal{F}$ and $\mathcal{F}^{-1}$ are Fourier and Inverse Fourier transform respectively. The key insight in FNO is that instead of parametrizing the kernel $k_\theta$, we parametrize its Fourier transform $\mathcal{F}(k_\theta)$ directly. That is, we parametrize the kernel transform operator as

$$\mathcal{K}_\beta v = \mathcal{F}^{-1}\Big( \Lambda_\beta\, \mathcal{F}(v) \Big).$$

This is a linear operator and will be referred to as *Fourier linear operator*. When $|\Lambda_\beta(\cdot)|_{\ell^1} < \infty$, we can write this

$$(\mathcal{K}_\beta v)(y) = \sum_{m \in \mathbb{Z}^d} e^{2\pi\, \mathrm{i}\langle m, y\rangle}\, \Lambda_\beta(m)\, (\mathcal{F}v)(m) \qquad \forall y \in \mathcal{Y}.$$

There are two practical challenges in implementing the operator $\mathcal{K}_\beta$. First, the implementation involves an infinite sum over $\mathbb{Z}^d$. Second, the Fourier transform $\mathcal{F}v$ cannot be computed exactly since the function $v$ is

only available on a finite grid of domain points. To address the first challenge, a large $K \in \mathbb{N}$ is fixed and we sum only over $m \in \mathbb{Z}^d$ such that $|m|_{\ell^\infty} \leq K$. The second challenge is addressed by approximating $\mathcal{F}v$ using the Discrete Fourier Transform (DFT) of $v$ over the finite grid of domain points, which can be efficiently computed using Fast Fourier Transform (FFT) algorithms. The solution to the second challenge motivates our DFT-based least-squares estimator.

## 1.3 Our Contribution

In this work, we study the error bounds of learning the operator class $\{v \mapsto \mathcal{F}^{-1}\big(\Lambda_\beta\,\mathcal{F}(v)\big) : \beta \in \mathcal{B}\}$, where $\mathcal{B}$ is some parameter space that will be specified later. We study this simple setup to conceptually separate the paradigm of operator learning from its commonly used instantiation using neural network architectures. By eliminating the complexities associated with neural networks, studying this linear class can provide insights that are broadly applicable to both algorithm design and theoretical analysis.

We assume that $\mathcal{V} = \mathcal{W} = \mathcal{H}^s(\mathbb{T}^d, \mathbb{R})$, a $(s, 2)$-Sobolev space of real-valued functions defined on the $d$-dimensional periodic torus. See Section 3.3 for an explanation on why $\mathcal{V}$ and $\mathcal{W}$ need to be function space with higher-order smoothness to achieve a vanishing error. We work in the agnostic (misspecified) setting and analyze the DFT-based least-squares estimator (see Section 3.2 for more details). Specifically, for some universal constant $c_1 > 0$, we show that the excess risk of the DFT-based least-squares estimator is at most

$$c_1 \left( \frac{1}{\sqrt{n}} + \frac{1}{N^s} + \frac{1}{K^{2s}} \right).$$

The term $1/\sqrt{n}$ is the usual *statistical/estimation* error due to a finite sample size. The term $1/K^{2s}$ is the *truncation* error incurred because the learner only works with the low Fourier modes $m$ such that $|m|_{\ell^\infty} \leq K$. Finally, the term $1/N^s$ is the *discretization* error due to functions being accessible to the learner only on the uniform grid of size $N^d$ of $[0, 1]^d$. This error quantifies the generalization error of an estimator trained on a grid of size $N^d$ but evaluated at full resolution ($N \to \infty$). It formalizes the concept of *multiresolution generalization* (operators trained at lower resolution have good generalization even when evaluated in higher resolution)–a phenomenon frequently observed in practice (Li et al., 2021, Section 5).

Additionally, we establish the lower bound on excess risk, showing that it is at least

$$c_2 \left( \frac{1}{n} + \frac{1}{N^{2s}} + \frac{1}{K^{2s}} \right)$$

for some $c_2 > 0$. Our analysis is non-asymptotic and the precise form of the constants $c_1$ and $c_2$ are provided in Theorems 3.2 and 3.3 respectively.

## 1.4 Related Works

After Li et al. (2021) proposed Fourier Neural Operators (FNOs), there has been a surge of interest in this architecture. The number of applied works is too vast and not entirely relevant to list here, so we focus on related theoretical works. One of the earliest theoretical analyses of FNOs was the universal approximation result by Kovachki et al. (2021).

More closely related to our work is a recent study on the sample complexity of various operator classes, including FNOs, by Kovachki et al. (2024a). Their scope is broader than ours as they address a general class of nonlinear operators. However, their results do not imply ours. They treat the truncation parameter $K$ as a part of the model rather than a variable that the learning algorithm can choose. Their error bounds are based on metric entropy analysis, which leads to a suboptimal dependence on $K$ and the input dimension $d$. Specifically, their bounds break down as $K \to \infty$ and suffer from the curse of dimensionality in $d$. In contrast, our work establishes statistical error bounds using sharp Rademacher analysis, avoiding both dependence on $K$ and the curse of dimensionality in $d$. An interesting future direction is to extend our Rademacher-based analysis to capture function classes at the level of generality considered in Kovachki et al. (2024a). We also note that Rademacher-based analysis has also been used by Raman et al. (2024); Tabaghi et al.

(2019) to study Schatten operators between Hilbert spaces. Kim & Kang (2024) also bound the Rademacher complexity of FNOs, but the bound is rather loose and even non-vanishing in some cases. Finally, the analysis by Liu et al. (2024) and Liu et al. (2025) also share our motivation of quantifying the statistical error in operator learning.

A recent work by Lanthaler et al. (2024) aligns with our goal of quantifying the discretization error of FNOs. In fact, the key ideas used in the proof of lemmas D.5 and D.6 in discretization error analysis is drawn from their work. However, the nature of their results differs from ours. To discuss the difference precisely, let $\Psi$ be a trained Fourier Neural Operator and $v$ be an input function available to the learner only over a discrete grid of domain points of size $N$. Denote $v^N$ as the set of discrete values of $v$ available to the learner. Lanthaler et al. (2024) bound the term $\|\Psi v - \Psi v^N\|$, quantifying the error incurred in the forward pass due to the function being available only over a discrete grid. Essentially, this only captures errors incurred during the test time but does not quantify the discretization error incurred during training. In contrast, our focus is on quantifying the generalization error of an operator trained on a grid of size $N^d$ but evaluated at full resolution ($N \to \infty$), a type of multiresolution generalization (Li et al., 2021, Section 5).

Finally, we also note that or setup is closely related to the function-to-function regression often studied in the functional data analysis (FDA) literature. For example, the linear layer of a neural operator $v \mapsto \mathcal{K}v + b$ is a well-studied model in FDA (Wang et al., 2016, Equation 15). Even a single layer of a neural operator $v \mapsto \sigma(\mathcal{K}v + b)$ has been examined in FDA literature as multi-index functional models (Wang et al., 2016, Equation 13), (Chen et al., 2011). That said, the overall goal of the FDA differs slightly from that of operator learning. In FDA, the focus is on statistical inference, typically using RKHS-based frameworks under some assumptions about the data-generating process. As a result, FDA methods often do not always scale to large datasets. In contrast, operator learning primarily aims at prediction, seeking to develop surrogate models that approximate numerical PDE solvers (Li et al., 2021; Kovachki et al., 2024b). The emphasis is on creating computationally efficient methods that can be used to train large models and handle large datasets. However, we believe that the intersection of these two fields can benefit both. The theoretical tools developed in FDA literature can be applied to the analysis of operator learning methods, while the computational advances in operator learning can help scale FDA methods.

## 2 Preliminaries

### 2.1 Notation

Let $\mathbb{N}$ be natural numbers and $\mathbb{Z}$ be integers. Define $\mathbb{N}_0 := \mathbb{N} \cup \{0\}$. $\mathbb{R}$ and $\mathbb{C}$ denote real and complex numbers respectively. For any $\eta \in \mathbb{R}^d$, we let $|\eta|_\infty := \max_{1 \le i \le d} |\eta_i|$ denote the $\ell^\infty$ norm. For any complex number $z \in \mathbb{C}$ such that $z = a + b\,\mathrm{i}$, we use $|z| = \sqrt{a^2 + b^2}$ and $\bar{z} = a - b\,\mathrm{i}$ denotes complex conjugate. For any $x, y \in \mathbb{R}^d$, the term $\langle x, y \rangle$ denotes the Euclidean inner product. Occasionally, the inner products on other Hilbert spaces such as $L^2$ will be distinguished from the Euclidean one with the subscript such as $\langle \cdot, \cdot \rangle_{L^2}$. However, when the context is clear, we will use $\langle \cdot, \cdot \rangle$ to denote canonical inner products on the respective Hilbert spaces.

Given $K \in \mathbb{N}$, we define $\mathbb{Z}^d_{\le K} = \{m \in \mathbb{Z}^d : |m|_\infty \le K\}$ and $\mathbb{Z}^d_{>K} := \mathbb{Z}^d \backslash \mathbb{Z}^d_{\le K}$. For a sequence $s := \{s_k\}_{k \in \mathbb{Z}^d}$, we will also use $|s|_{\ell^p}$ to denote the $\ell^p$ norm of $s$. Moreover, we let $\mathbb{T}^d \simeq [0,1]^d$ denote a $d$-dimensional periodic torus. See (Grafakos et al., 2008, Chapter 3) for more details on the torus. Throughout the paper, for any $m \in \mathbb{Z}^d$, we use $\varphi_m : \mathbb{T}^d \to \mathbb{C}$ to denote the function $\varphi_m(x) = e^{2\pi \mathrm{i} \langle m, x \rangle}$. The sequence $\{\varphi_m\}_{m \in \mathbb{Z}^d}$ will be referred to as Fourier basis.

### 2.2 $L^2$-Spaces and Fourier Analysis

Define

$$L^2(\mathbb{T}^d, \mathbb{R}) := \left\{ u : \mathbb{T}^d \to \mathbb{R} \mid \int_{\mathbb{T}^d} |u(x)|^2 \, dx < \infty \right\}.$$

Recall that $L^2(\mathbb{T}^d, \mathbb{R})$ is a Hilbert space with inner-product $\langle u, v \rangle_{L^2} = \int_{\mathbb{T}^d} u(x)\,v(x)\,dx$. The norm induced by this inner product will be denoted as $\|\cdot\|_{L^2}$. The sequence $\{\varphi_m\}_{m \in \mathbb{Z}^d}$ forms an orthonormal basis for

$L^2(\mathbb{T}^d, \mathbb{R})$. That is, for any $u \in L^2(\mathbb{T}^d, \mathbb{R})$, we can write $u = \sum_{m \in \mathbb{Z}^d} \langle u, \varphi_m \rangle_{L^2} \varphi_m$, where the convergence is in $L^2$-norm. The celebrated Parseval's identity then implies that $\|u\|_{L^2}^2 = \sum_{m \in \mathbb{Z}^d} |\langle u, \varphi_m \rangle_{L^2}|^2$.

Since $\mathbb{T}^d$ is identified with a bounded set $[0, 1]^d$, the condition $u \in L^2(\mathbb{T}^d, \mathbb{R})$ implies that $u$ is integrable. That is, $\int_{\mathbb{T}^d} |u(x)| \, dx < \infty$. For integrable functions, $\mathcal{F}$ denotes the Fourier transform operator such that $\mathcal{F}u : \mathbb{Z}^d \to \mathbb{C}$ is a complex-valued function on $\mathbb{Z}^d$ defined as

$$(\mathcal{F}u)(m) = \int_{\mathbb{T}^d} u(x) \, e^{-2\pi \, \mathrm{i} \langle m, x \rangle} \, dx.$$

Note that we have $(\mathcal{F}u)(m) = \langle u, \varphi_m \rangle$. We let $\mathcal{F}^{-1}$ denote the operator that satisfies $\left( \mathcal{F}^{-1} \mathcal{F} \right)(u) = u$. $\mathcal{F}^{-1}$ will be referred to as inverse Fourier transform. Note that even when $u$ is a real-valued function, $\langle u, \varphi_m \rangle$ may lie in $\mathbb{C}$.

### 2.3 Sobolev Spaces

Fix $s \in \mathbb{N}$ and define

$$\mathcal{H}^s(\mathbb{T}^d, \mathbb{R}) = \left\{ u \in L^2 \, \middle| \, \partial^k u \in L^2(\mathbb{T}^d, \mathbb{R}) \text{ for all } k \in \mathbb{N}_0^d \, \& \, |k|_\infty \leq s \right\}.$$

Here, $\partial^k u$ is the $k^{th}$ partial derivatives. The space $\mathcal{H}^s(\mathbb{T}^d, \mathbb{R})$, also referred to as $(s, 2)$-Sobolev space, is a Hilbert space with an inner product

$$\langle u, v \rangle_{\mathcal{H}^s} := \sum_{k \in \mathbb{N}_0^d \, : \, |k|_\infty \leq s} \left\langle \partial^k u, \partial^k v \right\rangle_{L^2},$$

which naturally induces the norm $\|u\|_{\mathcal{H}^s} := \sqrt{\langle u, u \rangle_{\mathcal{H}^s}}$. In this paper, we often assume that $s > d/2$. This ensures that (see Lemma D.4) $\sum_{m \in \mathbb{Z}^d} |\langle u, \varphi_m \rangle| < \infty$, which implies uniform convergence of the Fourier series over $\mathbb{T}^d$.

Note that it is more common to define Sobolev spaces with multi-indices $k$ such that $|k|_1 \leq s$. We chose the restriction $|k|_\infty \leq s$ simply for the convenience of computation. However, as $d$ is finite and all $\ell_p$ norms on a $d$-dimensional space are equivalent up to a factor of $d$.

## 3 Learning Fourier Linear Operators

In this section, we establish excess risk bounds of learning the operator class $\{v \mapsto \mathcal{F}^{-1}\left( \Lambda_\beta \, \mathcal{F}(v) \right) : \beta \in \mathcal{B}\}$, where $\mathcal{B}$ is some parameter space. Here, we only consider the case where $\mathcal{V}, \mathcal{W} \subseteq L^2(\mathbb{T}^d, \mathbb{R})$. This is different from the usual setting in the literature, where $\mathcal{V}$ and $\mathcal{W}$ are Banach spaces of vector-valued functions. First, a significant number of PDEs of practical interest describe how scalar-valued functions evolve. Since not much is known from a theoretical standpoint even for scalar-valued functions, we believe that this is a good start. Second, assuming $\mathcal{V}, \mathcal{W}$ to be a subset of $L^2$ (a Hilbert space) does not result in any meaningful loss of generality from a practical standpoint. In practice, one must discretize the domain and work with function values over a discrete grid, which effectively requires a bounded domain. This essentially means working with bounded functions on a bounded domain, all of which are $L^2$ integrable.

For scalar-valued functions, $\Lambda_\beta$ is a scalar-valued function defined on modes $\mathbb{Z}^d$. Since the function is only defined on a countable domain, we can also represent it by a scalar-valued sequence $\{\Lambda_\beta(m)\}_{m \in \mathbb{Z}^d}$. Henceforth, we will drop the $\beta$ and just write $\{\lambda_m\}_{m \in \mathbb{Z}^d}$, denoting $\lambda_m$'s to be the parameters themselves. For the convenience of notation, we will also use $\lambda$ to denote the sequence $\{\lambda_m\}_{m \in \mathbb{Z}^d}$ and write $\mathcal{F}^{-1}\left( \lambda \, \mathcal{F}(\cdot) \right)$. Fixing some $C > 0$, the class of interest can be written as

$$\left\{ v \mapsto \mathcal{F}^{-1}\left( \lambda \, \mathcal{F}(v) \right) : |\lambda|_{\ell^1} \leq C \right\}.$$

A starting point of our work is the following result on the decomposition of Fourier linear operators.

**Proposition 3.1.** *If $\lambda \in \ell^1(\mathbb{Z}^d)$, then*

$$\mathcal{F}^{-1}\big(\lambda \, \mathcal{F}(u)\big) = \sum_{m \in \mathbb{Z}^d} \lambda_m \, \varphi_m \, \langle \varphi_{-m}, u \rangle_{L^2}, \tag{1}$$

*where the equality holds for every $u \in L^2(\mathbb{T}^d, \mathbb{R})$.*

Here, $\varphi_m \otimes \varphi_{-m}$ is a rank-1 operator such that $(\varphi_m \otimes \varphi_{-m})(u) = \langle \varphi_{-m}, u \rangle_{L^2} \, \varphi_m$. The equality in (1) means $\mathcal{F}^{-1}\big(\lambda \, \mathcal{F}(u)\big) = \sum_{m \in \mathbb{Z}^d} \lambda_m \, \varphi_m \, \langle \varphi_{-m}, u \rangle_{L^2}$ for all $u \in L^2(\mathbb{T}^d, \mathbb{R})$, where the sum converges uniformly over $x \in \mathbb{T}^d$. We provide the proof of Proposition 3.1 in Appendix C.

Given Proposition 3.1, we can write our class as $\big\{ \sum_{m \in \mathbb{Z}^d} \lambda_m \, \varphi_m \otimes \varphi_{-m} : |\lambda|_{\ell^1} \leq C \big\}$. This representation is preferable for the following reasons. First, it highlights the fact that the Fourier basis is just one of the design choices for singular vectors that may be replaced with any other orthonormal sequences. Second, this representation also allows us to drop the constraint that $\lambda \in \ell^1$, which is a rather artificial constraint required only to ensure that the operator $\mathcal{F}^{-1}\big(\lambda \, \mathcal{F}(\cdot)\big)$ is a well-defined object. However, $\sum_{m \in \mathbb{Z}^d} \lambda_m \, \varphi_m \otimes \varphi_{-m}$ is still well-defined even when $\lambda \in \ell^\infty$ (in fact, it is a bounded operator). Therefore, for some fixed $C > 0$, we will instead study the class of operators

$$\mathcal{T} := \left\{ \sum_{m \in \mathbb{Z}^d} \lambda_m \, \varphi_m \otimes \varphi_{-m} \ \bigg| \ |\lambda|_{\ell^\infty} \leq C \right\}.$$

Since the class $\big\{ v \mapsto \mathcal{F}^{-1}\big(\lambda \, \mathcal{F}(\cdot)\big) : |\lambda|_{\ell^1} \leq C \big\}$ is contained in the class $\mathcal{T}$, any guarantee (in terms of upper bound) for $\mathcal{T}$ also holds for the $\ell^1$ constrained class.

**Remark.** The class $\mathcal{T}$ should remind readers of de Hoop et al. (2023), who also consider the problem of singular value inference of an operator under fixed singular vectors. However, their setting differs from ours in two significant ways. First, they only consider the well-specified setting with an additive noise model, whereas we adopt a fully agnostic viewpoint. Second, they do not account for possible discretization errors, assuming that their input and output functions are fully available to the learner.

## 3.1 Problem Setting and Error Types

We adopt the framework of statistical learning and study the rates of error in learning the class $\mathcal{T}$. In statistical learning, the learner is provided with $n \in \mathbb{N}$ i.i.d samples $S_n = \{(v_i, w_i)\}_{i=1}^n$ from some unknown distribution $\mu$ on $\mathcal{V} \times \mathcal{W}$. We adopt a fully agnostic viewpoint and do not make any assumptions about the data-generating process. Next, using the sample $S_n$ and some prespecified learning rule, the learner then finds an estimator $\widehat{T} \in \mathcal{T}$. We will abuse notation and denote $\widehat{T}$ to be both the learning rule and the estimator output by the learner. For an estimator $\widehat{T}$, we can define its expected excess risk as

$$\mathcal{E}_n(\widehat{T}, \mathcal{T}, \mu) = \underset{S_n \sim \mu^n}{\mathbb{E}} \left[ \underset{(v,w) \sim \mu}{\mathbb{E}} \left[ \|\widehat{T}v - w\|_{L^2}^2 \right] - \inf_{T \in \mathcal{T}} \underset{(v,w) \sim \mu}{\mathbb{E}} \left[ \|Tv - w\|_{L^2}^2 \right] \right].$$

Formally, the goal of the learner is to output the estimator such that $\mathcal{E}_n(\widehat{T}, \mathcal{T}, \mu) \to 0$ as $n \to \infty$. In traditional settings, the excess risk $\mathcal{E}_n(\widehat{T}, \mathcal{T}, \mu)$ is usually referred to as the statistical error of the learner. This error arises because the learner is trying to find the optimal operator in $\mathcal{T}$ for distribution $\mu$ while only having access to finitely many samples from the distribution. However, unlike traditional statistical learning settings, in operator learning, there are two additional errors beyond the statistical error: discretization error and truncation Error.

**Discretization Error:** The discretization error arises because the learner only has access to $(v_i, w_i) \sim \mu$ over some discrete grid of domain points. In this work, we assume that each $v_i$ and $w_i$ are available on a uniform grid

$$\mathrm{G} := \big\{ m/N : m \in \{0, \ldots, N-1\}^d \big\}$$

of $[0,1]^d$ for some prespecified $N \in \mathbb{N}$. That is, the learner only has access to $\{v_i(x) : x \in \mathrm{G}\}$ and $\{w_i(x) : x \in \mathrm{G}\}$. Although other grids are also used in practice, the use of FNO requires uniform griding. This is because the main benefit of FNO is its computationally efficient approximation of Fourier transform through fast Fourier transform (FFT) algorithms, which requires uniform grids.

**Truncation Error:** To see where the truncation error comes from, note that the representation of any estimator $T \in \mathcal{T}$ requires specifying an infinite sequence $\{\lambda_m\}_{m \in \mathbb{Z}^d}$. However, the infinite sequence cannot be implemented in a computer. Thus, for a practical implementation (Li et al., 2021), one picks a large $K \in \mathbb{N}$ and specifies the finite rank operator

$$T_K = \sum_{m \in \mathbb{Z}^d_{\leq K}} \lambda_m \, \varphi_m \otimes \varphi_{-m}.$$

While the truncation error is specific to our class of interest $\mathcal{T}$, a similar "truncation" error occurs in any model class. Such error arises because operator learning is inherently an infinite-dimensional problem, yet any computation we perform is limited to some finite-dimensional subspace.

### 3.1.1  Further Connection to FDA.

The operator $T_K$ is related to functional PCA-based estimators common in the FDA literature. Given $n$ i.i.d. function pairs $\{(v_i, w_i)\}_{i \leq n}$, the least-squares estimator solves $\sum_{i=1}^n w_i \otimes v_i = L \circ (\sum_{i=1}^n v_i \otimes v_i)$, which is under-specified in infinite-dimensional spaces. To address this, one computes a pseudo-inverse $(\sum_{i=1}^n v_i \otimes v_i)^\dagger$ by fixing an orthonormal basis $\{\psi_t\}_{t \in \mathbb{N}}$. With eigendecomposition $\sum_{i=1}^n v_i \otimes v_i = \sum_{t \geq 1} \eta_t \psi_t \otimes \psi_t$, the pseudo-inverse becomes $\sum_{t \geq 1} \mathbb{1}[\eta_t > 0] \eta_t^{-1} \psi_t \otimes \psi_t$, yielding the estimator $\widehat{L} = (\sum_{i=1}^n w_i \otimes v_i) \left(\sum_{t \geq 1} \mathbb{1}[\eta_t > 0] \, \eta_t^{-1} \psi_t \otimes \psi_t\right)$. In practice, the sum is truncated at some $t \leq \tau$.

Estimators of this type have been studied in works such as Hörmann & Kidziński (2015); Reimherr (2015); Yao et al. (2005) under well-specified models. These approaches generally require learning the basis functions $\psi_t$'s and the truncation parameter from the data to achieve the guarantees established in these studies, which often introduces significant computational challenges. In contrast, we work in the potentially misspecified (agnostic) setting, and $K$ depends only on the sample size $n$ to achieve $\sqrt{n}$-risk consistency. Additionally, FDA-based approaches typically assume exact access to the functions, which is unrealistic in practice. Instead, we explicitly account for the discretization error that arises when functions are only available on a finite grid.

### 3.2  A Constrained Least-Squares Estimator

In this section, we specify our primary estimator of interest. Let $T = \sum_{m \in \mathbb{Z}^d} \lambda_m \varphi_m \otimes \varphi_{-m}$. For any $v \in \mathcal{V}$, we have $Tv = \sum_{m \in \mathbb{Z}^d} \lambda_m \langle \varphi_{-m}, v \rangle \varphi_m$. As we only require $\ell^\infty$ norm of $\lambda$ to be bounded by $C$, we only get the convergence of the sum $\sum_{m \in \mathbb{Z}^d} \lambda_m \langle \varphi_{-m}, v \rangle \varphi_m$ in $L^2$ norm rather than uniform. Since $\{\varphi_m\}_{m \in \mathbb{Z}^d}$ is an orthonormal basis of $L^2(\mathbb{T}^d, \mathbb{R})$, Parseval's identity implies

$$\|Tv - w\|_{L^2}^2 = \sum_{m \in \mathbb{Z}^d} |\langle Tv - w, \varphi_m \rangle_{L^2}|^2$$

$$= \sum_{m \in \mathbb{Z}^d} |\lambda_m \langle \varphi_{-m}, v \rangle_{L^2} - \langle \varphi_{-m}, w \rangle_{L^2}|^2.$$

To see why the last equality is true, note that $\langle Tv, \varphi_m \rangle = \lambda_m \langle \varphi_{-m}, v \rangle$ and $\langle w, \varphi_m \rangle_{L^2} = \overline{\langle \varphi_m, w \rangle}_{L^2} = \langle \varphi_{-m}, w \rangle_{L^2}$ as $w$ is real-valued. Thus, given $\{(v_i, w_i)\}_{i=1}^n$, the least-squares estimator over the class $\mathcal{T}$ is an operator $T$ specified by the sequence $\{\lambda_m\}_{m \in \mathbb{Z}^d}$, which is obtained by solving the optimization problem

$$\min_{\{\lambda_m : m \in \mathbb{Z}^d\}} \frac{1}{n} \sum_{i=1}^n \sum_{m \in \mathbb{Z}^d} \left| \lambda_m \langle \varphi_{-m}, v_i \rangle_{L^2} - \langle \varphi_{-m}, w_i \rangle_{L^2} \right|^2 \quad \text{subject to} \quad \sup_{m \in \mathbb{Z}^d} |\lambda_m| \leq C.$$

However, this estimator cannot be implemented for two reasons. First, there is an infinite sum over $\mathbb{Z}^d$. Second the learner only has access to $(v_i, w_i)$ through $v_i^N := \{v_i(x) : x \in \mathrm{G}\}$ and $w_i^N := \{w_i(x) : x \in \mathrm{G}\}$, and thus the $L^2$ inner products cannot be computed exactly. Both of these issues can be resolved by considering the operator specified by the finite length sequence $\widehat{\lambda}(N) = \{\widehat{\lambda}_m : m \in \mathbb{Z}_{\leq K}^d\}$ obtained by minimizing

$$\frac{1}{n} \sum_{i=1}^n \sum_{m \in \mathbb{Z}_{\leq K}^d} \left| \lambda_m \, \mathrm{DFT}(v_i^N)(-m) - \mathrm{DFT}(w_i^N)(-m) \right|^2$$

subject to $\sup_{m \in \mathbb{Z}_{\leq K}^d} |\lambda_m| \leq C$. DFT, which stands for Discrete Fourier Transform, is the numerical approximation of $\langle \varphi_{-m}, u \rangle_{L^2}$ and is defined formally as

$$\mathrm{DFT}(u)(-m) := \frac{1}{N^d} \sum_{x \in \mathrm{G}} u(x) \, e^{-2\pi \mathrm{i} \langle x, m \rangle}.$$

To indicate the dependence of both truncation value $K$ and grid-size $N^d$, let us denote the estimator obtained by solving this problem to be $\widehat{T}_K^N$ where

$$\widehat{T}_K^N := \sum_{m \in \mathbb{Z}_{\leq K}^d} \widehat{\lambda}_m(N) \, \varphi_m \otimes \varphi_{-m}. \tag{2}$$

The estimator $\widehat{T}_K^N$ is the closest implementable version of the least-squares estimator for our setting.

### 3.3 Error Bounds

In this section, we study how $\mathcal{E}_n(\widehat{T}_K^N, \mathcal{T}, \mu)$ decay as a function of $n, K$ and $N$. Note that we have only specified that $\mathcal{V}$ and $\mathcal{W}$ are subsets of $L^2(\mathbb{T}^d, \mathbb{R})$, but have not specified their precise form. A natural choice would be $\mathcal{V} = \mathcal{W} = \{u \in L^2(\mathbb{T}^d, \mathbb{R}) : \|u\|_{L^2} \leq 1\}$, the unit ball of $L^2(\mathbb{T}^d, \mathbb{R})$. However, it turns out that $\mathcal{E}_n(\widehat{T}_K^N, \mathcal{T}, \mu)$ does not vanish under such $\mathcal{V}$ and $\mathcal{W}$.

To see this, let $K \in \mathbb{N}$ be a truncation parameter chosen by the learner. Define $\mu = \mathrm{Uniform}(\{(\psi_m, \psi_m) : 2^K < |m|_\infty < 2^{K+1}\})$ that is only supported on large modes. Here, $\psi_m = 2^{-1/2}(\varphi_m + \varphi_{-m})$ is the symmetrized, real-valued version of $m$-th Fourier mode. Note that we can choose a distribution as a function of $K$ because the truncation parameter $K$ can depend on the sample size $n$, but not on the exact realization of the samples.

For any sample size $n$ and the estimator $\widehat{T}_K^N$ produced by the learner, $\widehat{T}_K^N v = 0$ almost surely for $(v, w) \sim \mu$. Thus, we have $\mathbb{E}_{(v,w) \sim \mu}\left[\|\widehat{T}_K^N v - w\|_{L^2}^2\right] = \mathbb{E}_{(v,w) \sim \mu}\left[\|w\|_{L^2}^2\right] = 1$, as $w = \psi_m$ for some $2^K < |m|_\infty < 2^{K+1}$ almost surely and $\|\psi_m\|_{L^2} = 1$ for any $m \in \mathbb{Z}_{>0}^d$.

Next, let $C = 1$ and define $T^\star = \sum_{m \in \mathbb{Z}^d} \varphi_m \otimes \varphi_{-m}$. It is easy to see that $T^\star \psi_k = 2^{-\frac{1}{2}}(T^\star \varphi_k + T^\star \varphi_{-k}) = 2^{-\frac{1}{2}}(\varphi_{-k} + \varphi_k) = \psi_k \quad \forall k \in \mathbb{Z}^d \setminus \{\mathbf{0}\}$. As $T^\star \in \mathcal{T}$, we obtain $\inf_{T \in \mathcal{T}} \mathbb{E}_{(v,w) \sim \mu}\left[\|Tv - w\|_{L^2}^2\right] \leq \mathbb{E}_{(v,w) \sim \mu}\left[\|T^\star v - w\|_{L^2}^2\right] = 0$. Thus, we have established

$$\mathcal{E}_n(\widehat{T}_K^N, \mathcal{T}, \mu) \geq 1.$$

This shows that merely bounding the $L^2$ norm of $v, w$ is not sufficient to achieve a vanishing error. So, we need a stronger assumption on the input and output functions.

The inductive bias in FNOs is that the functions are sufficiently smooth so that the higher Fourier modes can be safely ignored. We will also adopt this viewpoint and assume that $\mathcal{V}$ and $\mathcal{W}$ are smooth subsets of $L^2(\mathbb{T}^d, \mathbb{R})$. In particular, we will assume that $\mathcal{V} = \mathcal{W} = \mathcal{H}^s(\mathbb{T}^d, \mathbb{R})$, a $(s, 2)$-Sobolev space (see Section 2.3). For any $u \in \mathcal{H}^s(\mathbb{T}^d, \mathbb{R})$, we are guaranteed that $\langle \varphi_{-m}, u \rangle_{L^2} \to 0$ sufficiently fast as $|m|_\infty \to \infty$. This allows us to ignore higher Fourier modes while only incurring small error. The following Theorem, whose proof is deferred to Apendix E, makes these arguments precise and provides an upper bound on the excess risk of $\widehat{T}_K^N$ in terms of $n, N$, and $K$.

**Theorem 3.2** (Upper Bound). *Let $\mathcal{V} = \mathcal{W} = \mathcal{H}^s(\mathbb{T}^d, \mathbb{R})$ for $s > d/2$ and $\mu$ be any distribution on $\mathcal{V} \times \mathcal{W}$ for which $\exists B > 0$ such that $\|v\|_{\mathcal{H}^s} \leq B$ and $\|w\|_{\mathcal{H}^s} \leq B$ almost surely. Then, for $n$ iid samples $\{(v_i, w_i)\}_{i=1}^n \sim \mu^n$ accessible to the learner over the $N$-uniform grid of $[0,1]^d$, the estimator $\widehat{T}_K^N$ defined in (2) for $N > \max\{5, 2K\}$ satisfies*

$$\mathcal{E}_n(\widehat{T}_K^N, \mathcal{T}, \mu) \leq 8B^2(C+1)^2 \left( \frac{1}{\sqrt{n}} + \frac{2^s \sqrt{\pi^d}}{N^s} + \frac{1}{K^{2s}} \right).$$

The terms $O(1/\sqrt{n})$, $O(1/N^s)$, and $O(1/K^{2s})$ are the estimator's statistical, discretization, and truncation errors respectively. For most practical applications of interest, we have $d = 3$ (functions defined on spatial coordinates). Since $\sqrt{\pi^d} \leq 6$ in these cases, the exponential dependence of the discretization error on $d$ is not an issue. Finally, choosing $N \geq n^{\frac{1}{2s}}$ and $K \geq n^{\frac{1}{4s}}$, Theorem 3.2 guarantees the $\sqrt{n}$– risk consistency of the estimator $\widehat{T}_K^N$.

**Proof Technique for Upper Bound:** Here, we highlight here the key technical novelties of our proof techniques and the implications of our results. To establish the upper bound, we first decompose the excess risk into three components: (1) the risk gap between the optimal operator in $\mathcal{T}$ for the distribution $\mu$ and its truncated counterpart, (2) the uniform deviation between the true empirical risk on the sample and its numerical approximation on the discrete grid, and (3) the uniform deviation between the empirical risk and the actual risk. This decomposition, introduced at the beginning of Appendix E, is not limited to the linear setting and can also be applied to analyze general non-linear operator classes. Given such decomposition, bounding the truncation error is straightforward using standard Fourier series properties for Sobolev spaces. The discretization error, however, requires nontrivial analysis to show that controlling the error of DFT suffices. Importantly, while the lower bound on the DFT error likely bounds the discretization error below, an upper bound on the DFT error does not always translate to an upper bound for the trained operator. For example, this is not true if one adds non-smooth activation such as RELU to our model. For statistical error, standard techniques yield a bound of $\sqrt{\frac{K^d}{n}}$, which does not allow taking $K \to \infty$. Our key contribution is a refined analysis that achieves a $\frac{1}{\sqrt{n}}$ bound independent of $K^d$. The $K$-independent bound is especially notable because $K$ in FNOs is analogous to the width in standard neural networks, where generalization bounds are known to be width-independent (Golowich et al., 2018). Our results provide initial evidence that similar $K$-free generalization bounds may be achievable for FNOs.

Our next result, proved in Appendix F, provides a lower bound on the rates at which $\mathcal{E}_n(\widehat{T}_K^N, \mathcal{T}, \mu)$ decay.

**Theorem 3.3** (Lower Bound). *Let $\mathcal{V} = \mathcal{W} = \mathcal{H}^s(\mathbb{T}^d, \mathbb{R})$ for $s > d/2$ and $C = 1$. Given $n, N, K \in \mathbb{N}$, there exists a distribution on $\mu$ on $\mathcal{V} \times \mathcal{W}$ for which $\exists B > 0$ such that $\|v\|_{\mathcal{H}^s} \leq B$ and $\|w\|_{\mathcal{H}^s} \leq B$ almost surely and for $n$ iid samples $\{(v_i, w_i)\}_{i=1}^n \sim \mu^n$ accessible over the $N$-uniform grid of $[0,1]^d$, the estimator $\widehat{T}_K^N$ defined in (2) for $N^s \geq \sqrt{2}B$ satisfies*

$$\mathcal{E}_n(\widehat{T}_K^N, \mathcal{T}, \mu) \geq \frac{B^2}{3(s+1)} \left( \frac{1}{8n} + \frac{1}{N^{2s}} + \frac{2}{(K+2)^{2s}} \right).$$

Although the lower bound on truncation error matches with the upper bound, there is a gap in the statistical and discretization error. We leave closing this gap for future work.

### 3.4 On Possible Extensions and Refinements of our Error Bounds

The smoothness assumptions in our work are primarily needed to control truncation and discretization errors. The lower bound in Section 3.3 shows that some regularity, specifically $s > 0$, is necessary for achieving a vanishing truncation error. This condition is also sufficient for our upper bound on the truncation error. The stronger requirement $s > d/2$ is required to ensure that the DFT-based estimator approximates the true Fourier coefficients. Moreover, even when $s = 0$, a statistical rate of $1/\sqrt{n}$ independent of $K$ can still be obtained under alternative assumptions. For example, if the operator's spectrum lies in $\ell^2(\mathbb{Z}^d)$, making it

Hilbert-Schmidt, results from Tabaghi et al. (2019); Raman et al. (2024) imply that such a rate is possible without any smoothness assumptions.

Additionally, in our analysis of the discretization error (Appendix E.2), the key quantity we control is the difference between the DFT approximation and the true Fourier coefficient, namely

$$|\operatorname{DFT}(u^N)(-m) - \langle \varphi_{-m}, u \rangle|.$$

The assumption of a uniform grid is used only to bound the numerical integration error introduced by the DFT. In principle, any numerical integration method can be applied to a non-uniform grid, and as long as its error vanishes with increasing grid resolution. For non-uniform grids with structure, such as those based on the roots of orthogonal polynomials, Gaussian quadrature rules may be used with standard accuracy guarantees. On unstructured grids, Monte Carlo methods with estimated importance weights can be used, although their convergence can be slow or the error may not vanish if the estimated weights have high variance.

## 4 Experiments

In this section, we present experiments demonstrating that our estimator achieves vanishing errors. We pick $d = 2$, and the input functions $v$ are sampled i.i.d. from $\mathcal{N}(0, 10^2(-\nabla^2 + \mathbf{I})^{-\gamma})$, a widely used distribution for generating training data in the operator learning literature (see Li et al. (2021); Kovachki et al. (2023)). Since $\gamma$ governs the decay rate of the eigenvalues of the covariance operator for this distribution, it directly controls the average smoothness of the samples $v$. For our experiments, we set $\gamma = 2$ as this is the smallest integer value that ensures $\gamma > d/2$ for $d = 2$.

To generate training data, we define a random operator

$$T^\star := \sum_{m \in \mathbb{Z}^d} \lambda_m \, \varphi_m \otimes \varphi_{-m},$$

where $\varphi_m$'s are Fourier modes and $\lambda_m \sim \mathrm{Unif}(-2, 2)$. For a given input $v$, the corresponding output is generated as $w = T^\star v + \varepsilon$, where $\varepsilon \sim \mathcal{N}(0, (-\nabla^2 + \mathbf{I})^{-3})$. Noise is sampled from a higher-order smooth space to ensure that its addition does not alter the smoothness of $w$. In actual implementation, the transformation $T^\star v$ is implemented on some $N \times N$ grid using Fast Fourier Transform (FFT) and Inverse Fast Fourier Transform (IFFT). The sum over $\mathbb{Z}^d$ is truncated at a Nyquist limit of $N/2$.

Recall that, our estimator in Section 3.2 is obtained by solving a convex optimization problem for $\lambda_m$'s for $m \in \mathbb{Z}^d_{\leq K}$. So, we implement the optimization routine for our estimator using stochastic gradient descent with a projection step to ensure $|\widehat{\lambda}_m| \leq 2$.

Figures 1, 2, and 3 show the statistical, truncation, and discretization errors, respectively. The $y$-axis in all these figures represents the relative mean-squared testing error:

$$\frac{1}{n_{\text{test}}} \sum_{i=1}^{n_{\text{test}}} \frac{\|w_i^{\text{true}} - w_i^{\text{predicted}}\|_{L^2}^2}{\|w_i^{\text{true}}\|_{L^2}},$$

evaluated using $n_{\text{test}} = 100$ i.i.d. samples. Additional experimental results are presented in Appendix G. The corresponding code is available at `https://github.com/unique-subedi/fourier-linear-operators`.

### 4.1 Statistical Error

Both training and testing are carried out on a $128 \times 128$ grid, with the estimator implemented using $K = 64$ modes. Error bands are included to account for fluctuations in the estimated parameters at small sample sizes, showing results from 5 independent runs. The model is trained and tested at the same resolution at the Nyquist limit of $K = 32$ modes to ensure that the reported error isolates statistical error with the minimum possible truncation and discretization errors. The smallest error is $\sim 6 \times 10^{-4}$ for the sample size of 500.

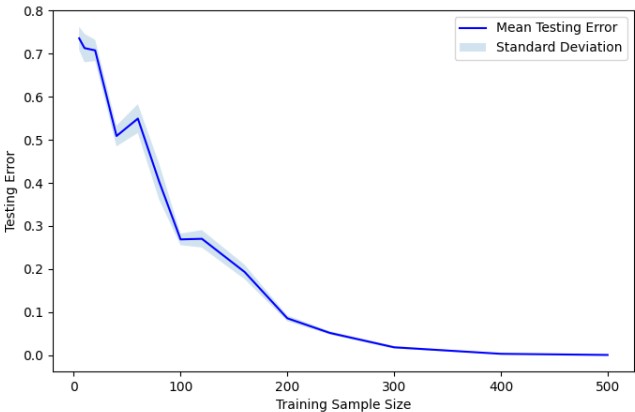

Figure 1: Statistical error of the estimator.

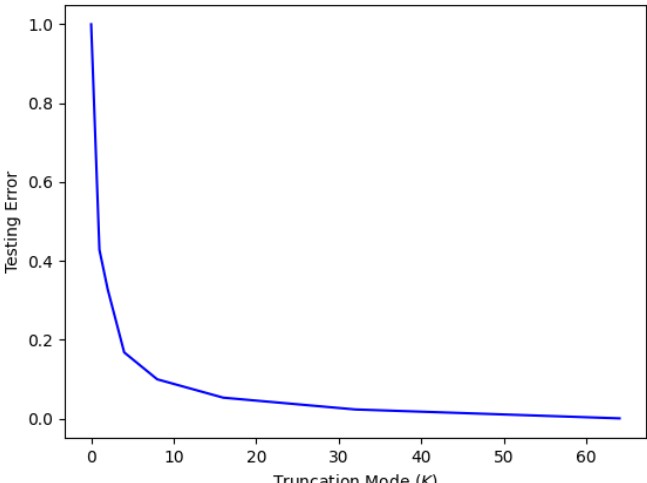

Figure 2: Truncation error of the estimator.

## 4.2 Truncation Error

Training and testing data are generated on a $128 \times 128$ grid, with the estimator trained using $n = 500$ samples. Error bands are omitted as the estimates are almost identical due to a large sample size. Both training and testing are conducted at the same resolution to remove discretization error, with the sample size selected to minimize statistical error, ensuring that the reported error isolates the truncation error effectively. The testing error converges to around $7.9 \times 10^{-4}$ at the Nyquist limit of $K = 64$.

## 4.3 Discretization Error

Testing data is generated on a $512 \times 512$ grid. The estimator is trained using $n = 500$ samples on grids of varying sizes $N \times N$, where $N \in \{1, 2, 4, 8, 16, 32, 64, 128, 256, 512\}$. For each training grid of size $N \times N$, truncation is performed at the Nyquist limit ($K = N/2$). The trained estimators are subsequently evaluated at the higher testing resolution of $512 \times 512$ to quantify discretization error. The testing error converges to around $6 \times 10^{-4}$ when the estimator is trained at a full grid size of $512 \times 512$ with 500 training samples.

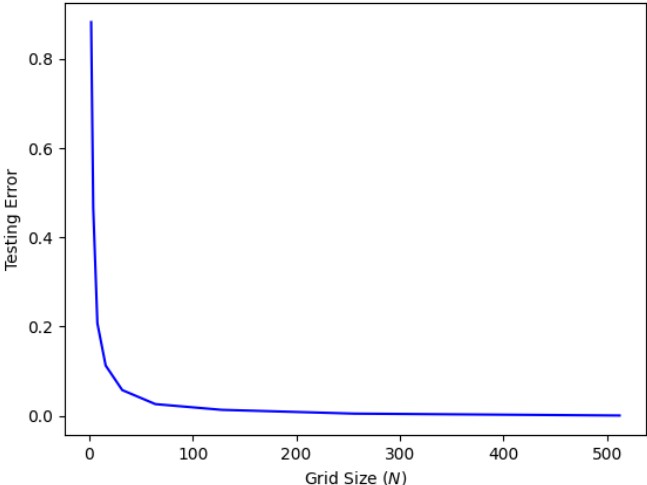

Figure 3: Discretization error of the estimator.

## 4.4   Summary of Experimental Findings

Our experiments in this section confirm that all three sources of error—statistical, truncation, and discretization—can be independently reduced by increasing the respective parameters $n$, $K$, and $N$.

However, the observed convergence rates, as reported in Appendix G, reveal some gaps compared to our theoretical predictions. In particular, the statistical error appears to decay faster than expected, and may depend on the smoothness parameter $\gamma$. This suggests that a refined, distribution-dependent analysis could yield sharper bounds beyond the worst-case setting. The largest discrepancy arises in the discretization error, where we observed almost uniform rate for all smoothness parameters. While our theory assumes the test resolution $N_2 \to \infty$, the experiments use a fixed resolution $N_2 = 512$ for computational reasons. This mismatch may account for the unexpectedly uniform decay rate across different smoothness levels. A more detailed analysis under finite resolutions could help explain this gap.

## 5   Discussion and Future Work

In this work, we established the excess risk error bounds of learning the core linear layer $v \mapsto \mathcal{F}^{-1}\big(\Lambda_\beta\,\mathcal{F}(v)\big)$ of Fourier neural operators. A natural future direction is to extend these results to single layer Fourier neural operator, $v \mapsto \sigma\big(\mathcal{F}^{-1}\big(\Lambda_\beta\,\mathcal{F}(v)\big) + b\big)$ and then to multiple layers. Although simple metric entropy-based analysis gives a bound on statistical error even for single layer neural operator, such a bound is vacuous when $K \to \infty$. It would be interesting to see if we can get a meaningful statistical rate even at the limit of $K \to \infty$. One can view $K$ as an analog of the width of traditional neural networks. Thus, analysis of $v \mapsto \sigma\big(\mathcal{F}^{-1}\big(\Lambda_\beta\,\mathcal{F}(v)\big) + b\big)$ as $K \to \infty$ can lead to a neural tangent kernel theory (Jacot et al., 2018) for operator learning. These insights will help us better understand width vs depth tradeoffs in operator learning.   One approach to extending our statistical rates to nonlinear operator could be to carry out a Rademacher analysis similar to that of Golowich et al. (2018) for finite-dimensional neural networks. The main technical challenge arises from the nature of the pointwise nonlinearity $\sigma$. In finite dimensions, for $v \in \mathbb{R}^p$, Golowich et al. (2018) exploit the identity $\sigma(v) = \sum_{j=1}^p \sigma(\langle v, e_j\rangle)e_j$, where $\{e_j\}$ is the standard basis. However, this identity no longer holds in the infinite-dimensional setting considered in Kovachki et al. (2023), where $\sigma$ is applied pointwise in the spatial domain rather than in the spectral domain.

For discretization error, we consider the setup where the training data is available on a grid of size $N^d$ but the trained operator is evaluated at full resolution ($N \to \infty$). It would be interesting to study the discretization error when the training data is available at resolution $N_1$, but the trained operator is evaluated at resolution

$N_2$. Such a theory would provide a more fine-grained quantification of multi-resolution generalization error observed in practices (Li et al., 2021). Additionally, a key practical limitation of our analysis is its reliance on uniform grids. As discussed in Section 3.4, the uniform grid assumption is used solely to bound the numerical integration error from the DFT. In principle, any integration scheme on a non-uniform grid could replace the DFT, as long as the approximation error vanishes with grid refinement. For structured grids (e.g., nodes from orthogonal polynomials), Gaussian quadrature can be used directly. For unstructured grids, one could apply Monte Carlo methods with importance weights, though these may exhibit slow convergence or non-vanishing bias when the weight variance is high. Extending our analysis to such general grids and formalize how discretization error interacts with other sources of error remains an interesting future direction.

Finally, with PDEs as an application, it is unclear if the iid-based statistical model is the right framework for operator learning. For instance, Boullé et al. (2023); Subedi & Tewari (2025) show that an active learning approach for data collection and training for solution operators of elliptic PDEs yields exponential error decay with increasing sample size. Therefore, an important future direction is to define the appropriate active learning model and develop active algorithms for operator learning.

## Acknowledgements

We acknowledge the support of NSF via grant DMS-2413089.

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

## A  Appendix

## B  Fourier Linear Operators

In this section, we provide a formal treatment of Fourier linear operators and the corresponding parametrization in FNOs. Recall that, in the Fourier Neural operator, one assumes that $\mathcal{X} = \mathcal{Y} = \mathbb{T}^d$ and the kernel is translation invariant. This implies that $\mathcal{K}_\theta$ defined in Section 1.1 is a convolution operator. That is,

$$\mathcal{K}_\theta \, v = k_\theta \star v, \quad \text{where} \quad (k_\theta \star v)(y) = \int_{\mathbb{T}^d} k_\theta(y - x) \, v(x) \, dx.$$

The convolution is done elementwise, $(\mathcal{K}_\theta v)_i(y) = \sum_{j=1}^p \big([k_\theta]_{ij} \star v_j\big)(y)$, where $[k_\theta]_{ij} : \mathbb{T}^d \to \mathbb{R}$ is the scalar-valued kernel defined by the $(i,j)^{th}$ component of $k_\theta$ and $(\mathcal{K}_\theta v)_i$ is the $i^{th}$ component of a $\mathbb{R}^q$-valued function. Similarly, $v_j : \mathbb{T}^d \to \mathbb{R}$ is the $j^{th}$ component function of $\mathbb{R}^p$-valued function $v$. Next, using the linearity of the Fourier transform and the Convolution Theorem, we can write

$$(\mathcal{K}_\theta v)_i = \mathcal{F}^{-1} \left( \sum_{j=1}^p \mathcal{F}\big([k_\theta]_{ij}\big) \, \mathcal{F}(v_j) \right).$$

where $\mathcal{F}$ is Fourier transform operator, and $\mathcal{F}^{-1}$ is the inverse Fourier transform. Here, $\mathcal{F}([k_\theta]_{ij}) : \mathbb{Z}^d \to \mathbb{C}$ and $\mathcal{F}(v_j) : \mathbb{Z}^d \to \mathbb{C}$ are Fourier transforms of $[k_\theta]_{ij}$ and $v_j$ respectively. Note that only discrete Fourier modes are defined because all the functions are defined on a periodic domain $\mathbb{T}^d$.

The key insight in FNO is that instead of parametrizing the kernel $k_\theta$, we parametrize its Fourier transform $\mathcal{F}(k_\theta)$ directly. That is, we parametrize the kernel transform operator as $(\mathcal{K}_\beta v)_i = \mathcal{F}^{-1} \left( \sum_{j=1}^p [\Lambda_\beta]_{ij} \, \mathcal{F}(v_j) \right)$ for some $\Lambda_\beta : \mathbb{Z}^d \to \mathbb{C}^{q \times p}$ that maps Fourier modes to a complex-valued matrix. Using the linearity of the inverse Fourier transform, we can write this more succinctly in a matrix form as $\mathcal{K}_\beta \, v = \mathcal{F}^{-1}\big(\Lambda_\beta \, \mathcal{F}(v)\big)$.

Since $\mathcal{F}^{-1}\big(\Lambda_\beta \, \mathcal{F}(v)\big)$ is a function defined on periodic domain $\mathbb{T}^d$, it has a Fourier series representation. So, we can write

$$\mathcal{F}^{-1}\big(\Lambda_\beta \, \mathcal{F}(v)\big)(\cdot) = \sum_{m \in \mathbb{Z}^d} \varphi_m(\cdot) \, \Lambda_\beta(m) \, (\mathcal{F}v)(m),$$

as $\varphi_m(\cdot) := e^{2\pi \, \mathrm{i} \langle m, \cdot \rangle}$ and the $m^{th}$ Fourier coefficient of $\mathcal{F}^{-1}\big(\Lambda_\beta \, \mathcal{F}(v)\big)$ is $\Lambda_\beta(m) \, (\mathcal{F}v)\,(m)$.

We have not specified in what metric the sum on the right-hand side converges. However, the convergence is not really an issue from a practical standpoint. In practice, $\Lambda_\beta$ is a trainable parameter, and it has been observed in Li et al. (2021) that parametrizing $\Lambda_\beta$ as a function from $\mathbb{Z}^d$ to $\mathbb{C}^{q \times p}$ yields sub-optimal results, possibly due to discrete structure of the lattice $\mathbb{Z}^d$. So, one picks a large $K > 0$ and parametrize $\Lambda_\beta$ as a collection of matrices $\{\Lambda_\beta(m) \, : \, m \in \mathbb{Z}^d$ such that $|m|_\infty \leq K\}$. In this case, the sum contains $\leq K^d$ terms and thus always converges. If one still wants to deal with the infinite sum, a standard assumption would be $[\Lambda_\beta]_{ij} \in \ell^1(\mathbb{Z}^d)$ for all $(i,j)$ pairs. That is, $\sum_{m \in \mathbb{Z}^d} |[\Lambda_\beta(m)]_{ij}| < \infty$ for all $(i,j)$ pairs. Then, the Weirstrass $M$-test implies that the sum above converges uniformly over all $y \in \mathbb{T}^d$.

Reparametrizing $\mathcal{K}_\theta$ as $\mathcal{F}^{-1}\big(\Lambda_\beta \, \mathcal{F}(v)\big)$ was proposed by Li et al. (2021) from the perspective of the convolution theorem, as discussed earlier. However, a more natural way to derive $\mathcal{F}^{-1}\big(\Lambda_\beta \, \mathcal{F}(v)\big)$ from $\mathcal{K}_\theta$ is to assume that $k_\theta$ has a Mercer-type decomposition.

**Proposition B.1.** *Let $k_\theta : \mathbb{Z}^d \to \mathbb{C}^{q \times p}$ be a kernel with decomposition*

$$[k_\theta(y, x)]_{ij} = \sum_{m \in \mathbb{Z}^d} [\Lambda_\beta(m)]_{ij} \, \varphi_m(y) \, \varphi_{-m}(x) \quad \forall (i,j)$$

*for some $\Lambda_\beta : \mathbb{Z}^d \to \mathbb{C}^{q \times p}$ such that $\Lambda_\beta \in \ell^1(\mathbb{Z}^d)$. Then, $\mathcal{K}_\theta v = \mathcal{F}^{-1}\big(\Lambda_\beta \, \mathcal{F}(v)\big)$ for all $v \in \mathcal{V}$.*

Given such decomposition, a simple algebra shows that $\int_{\mathbb{T}^d} [k_\theta(y, x)]_{ij} \, \varphi_k(x) dx = [\Lambda_\beta(k)]_{ij} \, \varphi_k(y)$. In other words, $[\Lambda_\beta(k)]_{ij}$ are the eigenvalues of the integral operator defined by the kernel $[k_\theta]_{ij}$. This suggests that

the Fourier layer of FNOs is parametrizing the eigenvalues of an operator while fixing the eigenfunctions to be $\varphi_k$'s. So, setting $\Lambda_\beta(m) = 0$ for $m \in \mathbb{Z}^d_{>K}$ amounts to parametrizing the low-rank version of such operator. This viewpoint shows that FNO is just a special case of a Low-rank Neural Operator defined in (Kovachki et al., 2023, Section 4.2).

More importantly, Proposition B.1 (see Appendix B.1 for the proof) provides a natural way to generalize Fourier Neural Operators. That is, we can consider $[k_\theta(y, x)]_{ij} = \sum_{m \in \mathcal{J}} [\Lambda_\beta(m)]_{ij} \, \psi_m(y) \, \phi_m(x)$, where $\mathcal{J}$ is some countable index-set and $\{\psi_m\}_{m \in \mathcal{J}}$, $\{\phi_m\}_{m \in \mathcal{J}}$ are some orthonormal sequences. Some common orthonormal sequences that allow efficient computation like FFT include the Chebyshev polynomial and wavelet basis. Some works have already explored the practical advantage of replacing Fourier basis with wavelet basis in certain problem settings Gupta et al. (2021); Tripura & Chakraborty (2023).

### B.1 Proof of Proposition B.1

We now end this section by proving Proposition B.1.

*Proof.* Let $\lambda_{ij}(m) := [\Lambda_\beta(m)]_{ij}$ and assume that

$$[k_\theta(y, x)]_{ij} = \sum_{m \in \mathbb{Z}^d} \lambda_{ij}(m) \, \varphi_m(y) \, \varphi_{-m}(x).$$

Using this decomposition, we obtain

$$(\mathcal{K}_\theta v)_i(y) = \int_{\mathbb{T}^d} \sum_{j=1}^p [k_\theta(y, x)]_{ij} \, v_j(x) \, dx$$

$$= \int_{\mathbb{T}^d} \sum_{j=1}^p \sum_{m \in \mathbb{Z}^d} \lambda_{ij}(m) \, \varphi_m(y) \, \varphi_{-m}(x) \, v_j(x) \, dx$$

$$= \sum_{m \in \mathbb{Z}^d} \varphi_m(y) \sum_{j=1}^p \lambda_{ij}(m) \int_{\mathbb{T}^d} \varphi_{-m}(x) \, v_j(x) \, dx.$$

Note that swapping the integral and the summation is justified through Fubini's because the sum over $\mathbb{Z}^d$ converges absolutely (as $\Lambda_\beta \in \ell^1$) and $\mathbb{T}^d$ is a bounded set. Since

$$\int_{\mathbb{T}^d} \varphi_{-m}(x) \, v_j(x) \, dx = \int_{\mathbb{T}^d} e^{-2\pi \mathrm{i} \langle m, x \rangle} v_j(x) \, dx = \mathcal{F}(v_j)(m),$$

we can write

$$(\mathcal{K}_\theta v)_i(y) = \sum_{m \in \mathbb{Z}^d} \varphi_m(y) \sum_{j=1}^p \lambda_{ij}(m) \, \mathcal{F}(v_j)(m).$$

Next, consider the function $w := \mathcal{F}^{-1}\left(\sum_{j=1}^p \lambda_{ij} \, \mathcal{F}(v_j)\right)$. Our proof will be complete upon showing that $w(y) = (\mathcal{K}_\theta v)_i(y)$ for every $y \in \mathbb{T}^d$. Since the function $w : \mathbb{T}^d \to \mathbb{C}$ is defined on a periodic domain, it has a Fourier series representation. That is,

$$w(y) = \sum_{m \in \mathbb{Z}^d} e^{2\pi \mathrm{i} \langle m, y \rangle} \mathcal{F}(w)(m) = \sum_{m \in \mathbb{Z}^d} e^{2\pi \mathrm{i} \langle m, y \rangle} \sum_{j=1}^p \lambda_{ij}(m) \, \mathcal{F}(v_j)(m),$$

where the final equality follows because $\mathcal{F}\left(\mathcal{F}^{-1}\left(\sum_{j=1}^p \lambda_{ij} \, \mathcal{F}(v_j)\right)\right)(m) = \sum_{j=1}^p \lambda_{ij}(m) \, \mathcal{F}(v_j)(m)$. As usual, $\Lambda_\beta \in \ell^1$ implies that the sum above converges uniformly over $y \in \mathbb{T}^d$. Recalling that $\varphi_m(y) = e^{2\pi \mathrm{i} \langle m, y \rangle}$, we have shown that $(\mathcal{K}_\theta v)_i(y) = w(y)$ for all $y \in \mathbb{T}^d$. This subsequently implies that

$$(\mathcal{K}_\theta v)_i = w = \mathcal{F}^{-1}\left(\sum_{j=1}^p \lambda_{ij} \, \mathcal{F}(v_j)\right).$$

Finally, using the linearity of the inverse Fourier transform and writing this in the matrix form establishes that $\mathcal{K}_\theta v = \mathcal{F}^{-1}(\Lambda_\beta \mathcal{F}(v))$ for any $v \in \mathcal{V}$. $\qquad \square$

## C  Proof of Proposition 3.1

*Proof.* Fix $v \in \mathcal{V}$ and define $w := \mathcal{F}^{-1}\big(\lambda \mathcal{F}(v)\big)$. By definition of the operator $\mathcal{F}^{-1}\big(\lambda \mathcal{F}(\cdot)\big)$, we have

$$w = \mathcal{F}^{-1}\left(\lambda \mathcal{F}(v)\right).$$

Using the Fourier series representation of $w$, we have

$$w(\cdot) = \sum_{m \in \mathbb{Z}^d} e^{2\pi \mathrm{i}\langle m, \cdot\rangle} \, (\mathcal{F}w)(m) = \sum_{m \in \mathbb{Z}^d} e^{2\pi \mathrm{i}\langle m, \cdot\rangle} \lambda_m \, \mathcal{F}(v)(m).$$

This step is rigorously justified because $\lambda \in \ell^1$. Noting that

$$(\mathcal{F}v)(m) = \int_{\mathbb{T}^d} e^{-2\pi \mathrm{i}\langle m, x\rangle} \, v(x) \, dx = \langle \varphi_{-m}, v\rangle_{L^2},$$

we can write

$$w(\cdot) = \sum_{m \in \mathbb{Z}^d} e^{2\pi \mathrm{i}\langle m, \cdot\rangle} \lambda_m \, \langle \varphi_{-m}, v\rangle_{L^2}.$$

Thus, $w = \sum_{m \in \mathbb{Z}^d} \lambda_m \, \langle \varphi_{-m}, v\rangle_{L^2} \, \varphi_m$, where the convergence is uniform over $\mathbb{T}^d$. This implies that

$$\mathcal{F}^{-1}\left(\lambda \mathcal{F}(v)\right) = \sum_{m \in \mathbb{Z}^d} \lambda_m \, \langle \varphi_{-m}, v\rangle_{L^2} \, \varphi_m.$$

Since this equality holds for every $v \in \mathcal{V}$, we have

$$\mathcal{F}^{-1}\left(\lambda \mathcal{F}(\cdot)\right) = \sum_{m \in \mathbb{Z}^d} \lambda_m \, \varphi_m \otimes \varphi_{-m}.$$

$\qquad \square$

## D  Technical Lemmas

In this section, we state and derive some technical Lemmas that we use to prove Theorems 3.2 and 3.3.

**Lemma D.1.** *For any $u \in \mathcal{H}^s(\mathbb{T}^d, \mathbb{R})$ , we have*

$$\left| \langle \varphi_{-m}, u\rangle_{L^2} \right| \le \frac{\|u\|_{\mathcal{H}^s}}{(2\pi)^s \, |m|_\infty^s} \qquad \forall m \in \mathbb{Z}^d \backslash \{\mathbf{0}\}.$$

*Proof.* Fix $m \in \mathbb{Z}^d \backslash \{\mathbf{0}\}$ and let $|m_j| = |m|_\infty = \max_{1 \le i \le d} |m_i|$. Clearly, $m_j \ne 0$. Integrating by parts $s$ times with respect to variable $x_j$ in $x = (x_1, \ldots, x_d)$, we obtain

$$\langle \varphi_{-m}, u\rangle = \int_{\mathbb{T}^d} u(x) e^{-2\pi \mathrm{i}\langle m, x\rangle} \, dx = (-1)^s \int_{\mathbb{T}^d} (\partial_j^s u)(x) \frac{e^{-2\pi \mathrm{i}\langle m, x\rangle}}{(-2\pi \mathrm{i}\, m_j)^s} \, dx = \left(\frac{1}{2\pi \mathrm{i}\, m_j}\right)^s \langle \varphi_{-m}, \partial_j^s u\rangle.$$

Here, all boundary terms vanish because $\mathbb{T}^d$ does not have a boundary ((Grafakos et al., 2008, Proof of Theorem 3.3.9)). Taking absolute value on both sides, we obtain that

$$|m_j|^s \, |\langle \varphi_{-m}, u\rangle| = (2\pi)^{-s} \, |\langle \varphi_{-m}, \partial_j^s u\rangle|$$

Finally, using the fact that $\left|\langle \varphi_{-m}, \partial_j^s u\rangle\right| \le \|u\|_{\mathcal{H}^s}$ completes our proof. $\qquad \square$

**Lemma D.2.** *For any $u \in \mathcal{H}^s(\mathbb{T}^d, \mathbb{R})$, we have*

$$\sum_{m \in \mathbb{Z}^d} (1 + |m|_\infty^{2s}) \, |\langle \varphi_{-m}, u \rangle|^2 \leq \|u\|_{\mathcal{H}^s}^2 .$$

*Proof.* Fix $m \in \mathbb{Z}^d \backslash \{\mathbf{0}\}$ and let $|m_j| = |m|_\infty = \max_{1 \leq i \leq d} |m_i|$. Clearly, $m_j \neq 0$. Integrating by parts $s$ times with respect to variable $x_j$ in $x = (x_1, \dots, x_d)$, we obtain

$$\langle \varphi_{-m}, u \rangle = \int_{\mathbb{T}^d} u(x) e^{-2\pi \mathrm{i} \langle m, x \rangle} \, dx = (-1)^s \int_{\mathbb{T}^d} (\partial_j^s u)(x) \frac{e^{-2\pi \mathrm{i} \langle m, x \rangle}}{(-2\pi \mathrm{i} \, m_j)^s} \, dx = \left( \frac{1}{2\pi \mathrm{i} \, m_j} \right)^s \langle \varphi_{-m}, \partial_j^s u \rangle .$$

Here, all boundary terms vanish because $\mathbb{T}^d$ does not have a boundary ((Grafakos et al., 2008, Proof of Theorem 3.3.9)). Taking absolute value on both sides, we obtain that

$$|m_j|^s \, |\langle \varphi_{-m}, u \rangle| = (2\pi)^{-s} \, |\langle \varphi_{-m}, \partial_j^s u \rangle|$$

Noting that $|m_j| = |m|_\infty$, squaring and summing over all $m \in \mathbb{Z}^d \backslash \{\mathbf{0}\}$ to get

$$\sum_{m \in \mathbb{Z}^d \backslash \{\mathbf{0}\}} |m|_\infty^{2s} \, |\langle \varphi_{-m}, u \rangle|^2 = (2\pi)^{-2s} \sum_{m \in \mathbb{Z}^d \backslash \{\mathbf{0}\}} |\langle \varphi_{-m}, \partial_j^s u \rangle|^2 \leq (2\pi)^{-2s} \left\| \partial_j^s u \right\|_{L^2}^2 ,$$

where the final inequality uses Parseval's identity and the fact that $\partial_j^s u \in L^2(\mathbb{T}^d, \mathbb{R})$. Thus, we obtain

$$\begin{aligned}
\sum_{m \in \mathbb{Z}^d} (1 + |m|_\infty^{2s}) \, |\langle \varphi_{-m}, u \rangle|^2 &= \sum_{m \in \mathbb{Z}^d} |\langle \varphi_{-m}, u \rangle|^2 + \sum_{m \in \mathbb{Z}^d \backslash \{\mathbf{0}\}} |m|_\infty^{2s} \, |\langle \varphi_{-m}, u \rangle|^2 \\
&\leq \|u\|_{L^2}^2 + (2\pi)^{-2s} \left\| \partial_j^s u \right\|_{L^2}^2 \\
&\leq \|u\|_{L^2}^2 + \left\| \partial_j^s u \right\|_{L^2}^2 \\
&\leq \|u\|_{\mathcal{H}^s}^2 ,
\end{aligned}$$

completing our proof. $\qquad\square$

**Lemma D.3.** *For any $u \in \mathcal{H}^s(\mathbb{T}^d, \mathbb{R})$ such that $s \geq 0$ and $K \in \mathbb{Z}_{>0}$, we have*

$$\sum_{m \in \mathbb{Z}_{>K}^d} |\langle \varphi_{-m}, u \rangle|^2 \leq \frac{\|u\|_{\mathcal{H}^s}^2}{K^{2s}}$$

*Proof.* Observe that

$$\begin{aligned}
\sum_{m \in \mathbb{Z}_{>K}^d} |\langle \varphi_{-m}, u \rangle|^2 &= \sum_{m \in \mathbb{Z}_{>K}^d} (1 + |m|_\infty^{2s}) \, |\langle \varphi_{-m}, u \rangle|^2 \, \frac{1}{(1 + |m|_\infty^{2s})} \\
&\leq \frac{1}{1 + K^{2s}} \sum_{m \in \mathbb{Z}_{>K}^d} (1 + |m|_\infty^{2s}) \, |\langle \varphi_{-m}, u \rangle|^2 \\
&\leq \frac{\|u\|_{\mathcal{H}^s}^2}{K^{2s}} ,
\end{aligned}$$

using Lemma D.2. $\qquad\square$

**Lemma D.4.** *For any $u \in \mathcal{H}^s(\mathbb{T}^d, \mathbb{R})$ such that $s > d/2$, we have*

$$\sum_{m \in \mathbb{Z}_{>K}^d} |\langle \varphi_{-m}, u \rangle| \leq \|u\|_{\mathcal{H}^s} \sqrt{\frac{3^d}{2s - d}} \, \frac{1}{\sqrt{K^{2s-d}}} , .$$

*Proof.* First, we use Cauchy-Schwarz to get

$$\sum_{m\in\mathbb{Z}^d_{>K}} |\langle\varphi_{-m},u\rangle| = \sqrt{\sum_{m\in\mathbb{Z}^d_{>K}} (1+|m|^{2s}_\infty)|\langle\varphi_{-m},u\rangle|^2}\sqrt{\sum_{m\in\mathbb{Z}^d_{>K}} \frac{1}{(1+|m|^{2s}_\infty)}}$$

Lemma D.3 implies that the first term is $\leq \|u\|_{\mathcal{H}^s}$. To bound the second term, note that for any $j\in\mathbb{N}$, we have $|\{m\in\mathbb{Z}^d : |m|_\infty = j\}| = 2(2j+1)^{d-1}$. This is because one of the entry of $m$ has to be $\pm j$ and other $d-1$ entries could be anything in $\{-j\ldots,-1,0,1,\ldots,j\}$. So,

$$\sum_{m\in\mathbb{Z}^d_{>K}} \frac{1}{(1+|m|^{2s}_\infty)} = \sum_{j>K} \frac{2(2j+1)^{d-1}}{(1+j^{2s})} \leq 3^d\sum_{j>K} \frac{1}{j^{2s-d+1}} \leq 3^d\int_K^\infty t^{-2s+d-1}\,dt = \frac{3^d}{2s-d}\frac{1}{K^{2s-d}},$$

for all $s > d/2$. Thus, overall, we obtain

$$\sum_{m\in\mathbb{Z}^d_{>K}} |\langle\varphi_{-m},u\rangle| \leq \|u\|_{\mathcal{H}^s}\sqrt{\frac{3^d}{2s-d}}\frac{1}{\sqrt{K^{2s-d}}},$$

completing our proof. □

**Lemma D.5.** *Let* $\mathrm{G} := \{j/N : j\in\{0,\ldots,N-1\}^d\}$ *be the $N$-uniform grid of $[0,1]^d$. Then, for any $m\in\mathbb{Z}^d_{<N}$, we have*

$$\frac{1}{N^d}\sum_{x\in\mathrm{G}} e^{2\pi\,\mathrm{i}\langle k-m,x\rangle} = \mathbb{1}[k\equiv m\,(\mathrm{mod}\,N)].$$

Here, we say $k\equiv m(\mathrm{mod}\,N)$ if $\exists\ell\in\mathbb{Z}^d$ such that $k = N\ell + m$.

*Proof.* We first prove it for $d=1$. For this case, we need to show that

$$\frac{1}{N}\sum_{j=0}^{N-1} e^{2\pi\,\mathrm{i}(k-m)\frac{j}{N}} = \mathbb{1}[k\equiv m(\mathrm{mod}\,N)].$$

First, consider the case where $k = \tau N + m$ for some $\tau\in\mathbb{Z}$. Then, $e^{2\pi\,\mathrm{i}(k-m)\frac{j}{N}} = e^{2\pi\,\mathrm{i}\tau j} = 1$ by Euler's identity. Thus, the overall sum must be 1. Next, assume that $k\not\equiv m\,(\mathrm{mod}\,N)$. Then, the geometric series formula implies that

$$\frac{1}{N}\sum_{j=0}^{N-1} e^{2\pi\,\mathrm{i}(k-m)\frac{j}{N}} = \frac{1}{N}\frac{1-e^{2\pi\,\mathrm{i}(k-m)j}}{1-e^{2\pi\,\mathrm{i}(k-m)\frac{j}{N}}} = 0.$$

Here, the final equality holds because $e^{2\pi\,\mathrm{i}(k-m)j} = 1$ by Euler's identity, whereas $e^{2\pi\,\mathrm{i}(k-m)\frac{j}{N}} \neq 1$ for every $j\in\{0,1\ldots,N-1\}$. This completes our proof for the case $d=1$.

Next, to prove it for general $d$, we write the sum as $d$-fold summation

$$\frac{1}{N^d}\sum_{x\in\mathrm{G}} e^{2\pi\,\mathrm{i}\langle k-m,x\rangle} = \frac{1}{N^d}\sum_{j_1=0}^{N-1}\cdots\sum_{j_d=0}^{N-1} e^{2\pi\,\mathrm{i}(k_1-m_1)\frac{j_1}{N}}\cdots e^{2\pi\,\mathrm{i}(k_d-m_d)\frac{j_d}{N}} = \prod_{t=1}^d \frac{1}{N}\sum_{j_t=0}^{N-1} e^{2\pi\,\mathrm{i}(k_t-m_t)\frac{j_t}{N}}.$$

Using the result of $d=1$ case for each term in the product, we have

$$\frac{1}{N^d}\sum_{x\in\mathrm{G}} e^{2\pi\,\mathrm{i}\langle k-m,x\rangle} = \prod_{t=1}^d \mathbb{1}[k_t\equiv m_t\,(\mathrm{mod}\,N)] = \mathbb{1}[k\equiv m\,(\mathrm{mod}\,N)].$$

□

**Lemma D.6.** *Let $u \in \mathcal{H}^s(\mathbb{T}^d, \mathbb{R})$ such that $\|u\|_{\mathcal{H}^s} \leq B$ and $u^N := \{u(x) : x \in \mathrm{G}\}$ be its values on the uniform grid $\mathrm{G}$. Then, for all $|m|_\infty < N$, we have*

$$|\mathrm{DFT}(u^N)(-m) - \langle \varphi_{-m}, u \rangle| \leq \left| \sum_{\ell \in \mathbb{Z}^d \setminus \{\mathbf{0}\}} \langle \varphi_{-(\ell N + m)}, u \rangle \right|.$$

*Proof.* Recall that

$$\mathrm{DFT}(u^N)(-m) = \frac{1}{N^d} \sum_{x \in \mathrm{G}} u(x) \, e^{-2\pi \mathrm{i} \langle m, x \rangle}.$$

Pick some $M > N$ and write

$$u(x) = \sum_{k \in \mathbb{Z}^d_{\leq M}} \langle \varphi_{-k}, u \rangle \, e^{2\pi \mathrm{i} \langle k, x \rangle} + \left( u(x) - \sum_{k \in \mathbb{Z}^d_{\leq M}} \langle \varphi_{-k}, u \rangle \, e^{2\pi \mathrm{i} \langle k, x \rangle} \right).$$

We can then write

$$\mathrm{DFT}(u^N)(-m)$$

$$= \frac{1}{N^d} \sum_{x \in \mathrm{G}} \left( \sum_{k \in \mathbb{Z}^d_{\leq M}} \langle \varphi_{-k}, u \rangle \, e^{2\pi \mathrm{i} \langle k, x \rangle} + \left( u(x) - \sum_{k \in \mathbb{Z}^d_{\leq M}} \langle \varphi_{-k}, u \rangle \, e^{2\pi \mathrm{i} \langle k, x \rangle} \right) \right) e^{-2\pi \mathrm{i} \langle m, x \rangle}$$

$$= \sum_{k \in \mathbb{Z}^d_{\leq M}} \langle \varphi_{-k}, u \rangle \left( \frac{1}{N^d} \sum_{x \in \mathrm{G}} e^{2\pi \mathrm{i} \langle k - m, x \rangle} \right) + \frac{1}{N^d} \sum_{x \in \mathrm{G}} \left( u(x) - \sum_{k \in \mathbb{Z}^d_{\leq M}} \langle \varphi_{-k}, u \rangle \, e^{2\pi \mathrm{i} \langle k, x \rangle} \right) e^{-2\pi \mathrm{i} \langle m, x \rangle}$$

$$= \sum_{k \in \mathbb{Z}^d_{\leq M}} \langle \varphi_{-k}, u \rangle \, \mathbb{1}[k \equiv m (\mathrm{mod}\ N)] + \frac{1}{N^d} \sum_{x \in \mathrm{G}} \left( u(x) - \sum_{k \in \mathbb{Z}^d_{\leq M}} \langle \varphi_{-k}, u \rangle \, e^{2\pi \mathrm{i} \langle k, x \rangle} \right) e^{-2\pi \mathrm{i} \langle m, x \rangle},$$

where the final equality follows from Lemma D.5 as $|m|_\infty < N$. Note that we can swap sums over $\mathrm{G}$ and $\mathbb{Z}^d$ in the first term because the sums converge absolutely when $s > d/2$ (see Lemma D.4). Thus, we obtain

$$|\mathrm{DFT}(u^N)(-m) - \langle \varphi_{-m}, u \rangle| \leq \left| \sum_{k \in \mathbb{Z}^d_{\leq M}} \langle \varphi_{-k}, u \rangle \, \mathbb{1}[k \equiv m (\mathrm{mod}\ N)] - \langle \varphi_{-m}, u \rangle \right|$$

$$+ \left| \frac{1}{N^d} \sum_{x \in \mathrm{G}} \left( u(x) - \sum_{k \in \mathbb{Z}^d_{\leq M}} \langle \varphi_{-k}, u \rangle \, e^{2\pi \mathrm{i} \langle k, x \rangle} \right) e^{-2\pi \mathrm{i} \langle m, x \rangle} \right|$$

Using the uniform bound over $x \in \mathrm{G}$ for the second term and the following identity for the first term

$$\sum_{k \in \mathbb{Z}^d_{\leq M}} \langle \varphi_{-k}, u \rangle \, \mathbb{1}[k \equiv m (\mathrm{mod}\ N)] - \langle \varphi_{-m}, u \rangle = \sum_{k \in \mathbb{Z}^d_{\leq M} \setminus \{m\}} \langle \varphi_{-k}, u \rangle \, \mathbb{1}[k \equiv m (\mathrm{mod}\ N)],$$

we obtain

$$|\mathrm{DFT}(u^N)(-m) - \langle \varphi_{-m}, u \rangle|$$

$$\leq \left| \sum_{k \in \mathbb{Z}^d_{\leq M} \setminus \{m\}} \langle \varphi_{-k}, u \rangle \, \mathbb{1}[k \equiv m (\mathrm{mod}\ N)] \right| + \sup_{x \in \mathrm{G}} \left| u(x) - \sum_{k \in \mathbb{Z}^d_{\leq M}} \langle \varphi_{-k}, u \rangle \, e^{2\pi \mathrm{i} \langle k, x \rangle} \right|$$

Recall that we have (i) $|\langle \varphi_{-k}, u \rangle\, e^{2\pi\, \mathrm{i}\langle k, x\rangle}| \leq B$ and $\sum_{k \in \mathbb{Z}^d} |\langle \varphi_{-k}, u\rangle\, e^{2\pi\, \mathrm{i}\langle k, x\rangle}| < \infty$ for $s > d/2$ using Lemma D.4. The Weierstrass M-test implies that the second term converges to 0 uniformly over $x \in \mathbb{T}^d$ as $M \to \infty$. Thus, we obtain

$$\sum_{k \in \mathbb{Z}^d_{\leq M} \setminus \{m\}} \langle \varphi_{-k}, u\rangle\, \mathbb{1}[k \equiv m(\mathrm{mod}\ N)] \xrightarrow[M \to \infty]{} \sum_{k \in \mathbb{Z}^d \setminus \{m\}} \langle \varphi_{-k}, u\rangle\, \mathbb{1}[k \equiv m(\mathrm{mod}\ N)]$$

$$= \sum_{\ell \in \mathbb{Z}^d \setminus \{\mathbf{0}\}} \langle \varphi_{-(\ell N + m)}, u\rangle,$$

which completes our proof.

**Lemma D.7.** *For any $s \in \mathbb{N}$ such that $s > d/2$, we have*

$$\sum_{k \in \mathbb{Z}^d \setminus \{\mathbf{0}\}} \frac{1}{|k|^{2s}_\infty} \leq \pi^2\, 3^{d-2}.$$

*Proof.* Recall that $|\{m \in \mathbb{Z}^d : |m|_\infty = j\}| = 2(2j+1)^{d-1}$. This is because one of the entry of $m$ has to be $\pm j$ and other $d-1$ entries could be anything in $\{-j \ldots, -1, 0, 1, \ldots, j\}$. Thus,

$$\sum_{\ell \in \mathbb{Z}^d \setminus \{\mathbf{0}\}} \frac{1}{|\ell|^{2s}_\infty} \leq \sum_{j=1}^\infty \frac{2(2j+1)^{d-1}}{j^{2s}} \leq 2 \cdot 3^{d-1} \sum_{j=1}^\infty \frac{1}{j^{2s-d+1}} \leq 2 \cdot 3^{d-1} \sum_{j=1}^\infty \frac{1}{j^2} = \frac{2 \cdot 3^{d-1} \pi^2}{6} = \pi^2\, 3^{d-2}.$$

The third inequality uses $2s - d \leq 1$ as $s > d/2$ and $s \in \mathbb{N}$. $\qquad\square$

$\qquad\square$

# E   Proof of Upper Bound (Theorem 3.2)

Before we prove Theorem 3.2, we need some notation. For any $T \in \mathcal{T}$ such that $T = \sum_{m \in \mathbb{Z}^d} \lambda_m\, \varphi_m \otimes \varphi_{-m}$, we define

$$r(T) := \mathop{\mathbb{E}}_{(v,w) \sim \mu} \left[ \|Tv - w\|^2_{L^2} \right] = \mathop{\mathbb{E}}_{(v,w) \sim \mu} \left[ \sum_{m \in \mathbb{Z}^d} |\lambda_m \langle \varphi_{-m}, v\rangle - \langle \varphi_{-m}, w\rangle|^2 \right]$$

$$\widehat{r}(T) := \frac{1}{n} \sum_{i=1}^n \|Tv_i - w_i\|^2_{L^2} = \frac{1}{n} \sum_{i=1}^n \sum_{m \in \mathbb{Z}^d} |\lambda_m \langle \varphi_{-m}, v_i\rangle - \langle \varphi_{-m}, w_i\rangle|^2$$

where $\{(v_i, w_i)\}_{i=1}^n$ is the sample accessible to the learner on a uniform grid of $[0,1]^d$. Then, using these definitions, we can write

$$\mathcal{E}_n(\widehat{T}^N_K, \mathcal{T}, \mu) = \mathbb{E}\left[ r(\widehat{T}^N_K) - \inf_{T \in \mathcal{T}} r(T) \right] = \mathbb{E}\left[ r(\widehat{T}^N_K) - \inf_{T \in \mathcal{T}_K} r(T) \right] + \inf_{T \in \mathcal{T}_K} r(T) - \inf_{T \in \mathcal{T}} r(T),$$

where $\mathcal{T}_K$ is the truncated class defined as

$$\mathcal{T}_K := \left\{ \sum_{m \in \mathbb{Z}^d_{\leq K}} \lambda_m\, \varphi_m \otimes \varphi_{-m} \;\middle|\; \sup_{m \in \mathbb{Z}^d_{\leq K}} |\lambda_m| \leq C \right\}.$$

Furthermore, defining

$$\widehat{T}_K \in \arg\min_{T \in \mathcal{T}_K} \widehat{r}(T),$$

we can decompose

$$\mathcal{E}_n(\widehat{T}_K^N, \mathcal{T}, \mu) = \underbrace{\mathbb{E}\left[r(\widehat{T}_K^N) - r(\widehat{T}_K)\right]}_{(\text{I})} + \underbrace{\mathbb{E}\left[r(\widehat{T}_K) - \inf_{T \in \mathcal{T}_K} r(T)\right]}_{(\text{II})} + \underbrace{\inf_{T \in \mathcal{T}_K} r(T) - \inf_{T \in \mathcal{T}} r(T)}_{(\text{III})}.$$

First, it is easy to see that

$$(\text{III}) \leq \sup_{T \in \mathcal{T}} \inf_{T_K \in \mathcal{T}_K} |r(T) - r(T_K)|.$$

To upper bound (II), let $T_K^\star \in \mathcal{T}_K$ such that $r(T_K^\star) = \inf_{T \in \mathcal{T}_K} r(T)$. Formally, for every $\varepsilon > 0$, we may only be guaranteed the existence of $T_K^\star$ such that $r(T_K^\star) \leq \inf_{T \in \mathcal{T}_K} r(T) + \varepsilon$. However, as $\varepsilon$ can be made arbitrarily small, we can just choose it to be smaller than any error bound we obtain at the end. So, the arguments below are rigorously justified.

Given such $T_K^\star$, we can write

$$(\text{II}) = \mathbb{E}[r(\widehat{T}_K) - r(T_K^\star)] = \mathbb{E}[r(\widehat{T}_K) - \widehat{r}(\widehat{T}_K)] + \mathbb{E}[\widehat{r}(\widehat{T}_K) - \widehat{r}(T_K^\star)] + \mathbb{E}[\widehat{r}(T_K^\star) - r(T_K^\star)].$$

The last term of the sum vanishes because $\mathbb{E}[\widehat{r}(T_K^\star)] = r(T_K^\star)$. As for the second term, $\widehat{T}_K$ minimizes empirical loss over the samples, implying $\widehat{r}(\widehat{T}_K) \leq \widehat{r}(T_K^\star)$. For the first term, we use the trivial bound $r(\widehat{T}_K) - \widehat{r}(\widehat{T}_K) \leq \sup_{T \in \mathcal{T}_K} |r(T) - \widehat{r}(T)|$. Overall, we obtain

$$(\text{II}) \leq \mathbb{E}\left[\sup_{T \in \mathcal{T}_K} |r(T) - \widehat{r}(T)|\right].$$

Finally, we upper bound the term (I). Given $K$ and $N$, for any $T \in \mathcal{T}_K$ such that $T = \sum_{m \in \mathbb{Z}_{\leq K}^d} \lambda_m \, \varphi_m \otimes \varphi_{-m}$, define

$$\widehat{r}_N(T) := \frac{1}{n} \sum_{i=1}^n \sum_{m \in \mathbb{Z}_{\leq K}^d} \left|\lambda_m \operatorname{DFT}(v_i^N)(-m) - \operatorname{DFT}(w_i^N)(-m)\right|^2 + \frac{1}{n} \sum_{i=1}^n \sum_{m \in \mathbb{Z}_{>K}^d} |\langle \varphi_{-m}, w_i \rangle|^2.$$

Technically, the term $\widehat{r}_N(T)$ also depends on $K$, but we drop $K$ to avoid cluttered notation. Here, the first term above is the empirical DFT-based least squares loss of $T$ define in 3.2. The second term is introduced purely for technical reasons to make our calculations work (see Section E.2). Since the second term does not depend on $T$, our estimator $\widehat{T}_K^N$ is still the operator obtained by minimizing $\widehat{r}_N$. Then, note that

$$(\text{I}) = \mathbb{E}[r(\widehat{T}_K^N) - \widehat{r}_N(\widehat{T}_K^N)] + \mathbb{E}[\widehat{r}_N(\widehat{T}_K^N) - \widehat{r}_N(\widehat{T}_K)] + \mathbb{E}[\widehat{r}_N(\widehat{T}_K) - r(\widehat{T}_K)]$$

Note that the second term above satisfies $\widehat{r}_N(\widehat{T}_K^N) - \widehat{r}_N(\widehat{T}_K) \leq 0$ almost surely because $\widehat{T}_K^N$ minimizes $\widehat{r}_N(T)$ over all $T \in \mathcal{T}_K$. For the first and the third term, we use the bound

$$\mathbb{E}[r(\widehat{T}_K^N) - \widehat{r}_N(\widehat{T}_K^N)] \leq \mathbb{E}[\sup_{T \in \mathcal{T}_K} |r(T) - \widehat{r}_N(T)|] \quad \text{and} \quad \mathbb{E}[\widehat{r}_N(\widehat{T}_K) - r(\widehat{T}_K)] \leq \mathbb{E}[\sup_{T \in \mathcal{T}_K} |r(T) - \widehat{r}_N(T)|].$$

Thus, we have

$$(\text{I}) \leq 2\,\mathbb{E}\left[\sup_{T \in \mathcal{T}_K} |r(T) - \widehat{r}_N(T)|\right] \leq 2\,\mathbb{E}\left[\sup_{T \in \mathcal{T}_K} |r(T) - \widehat{r}(T)|\right] + 2\,\mathbb{E}\left[\sup_{T \in \mathcal{T}_K} |\widehat{r}(T) - \widehat{r}_N(T)|\right],$$

where the final step uses the triangle inequality. Combining everything, we have established that

$$\mathcal{E}_n(\widehat{T}_K^N, \mathcal{T}, \mu) \leq 3\,\mathbb{E}\left[\sup_{T \in \mathcal{T}_K} |r(T) - \widehat{r}(T)|\right] + 2\,\mathbb{E}\left[\sup_{T \in \mathcal{T}_K} |\widehat{r}(T) - \widehat{r}_N(T)|\right] + \sup_{T \in \mathcal{T}} \inf_{T_K \in \mathcal{T}_K} |r(T) - r(T_K)|.$$

The first term is the statistical error, the second is the discretization error, and the final is the truncation error. Next, we bound each of these terms individually.

**E.1   Upper bound on the truncation error** $\sup_{T \in \mathcal{T}} \inf_{T_K \in \mathcal{T}_K} |r(T) - r(T_K)|$

Pick any $T \in \mathcal{T}$. Then, there exists a sequence $\{\lambda_m\}_{m \in \mathbb{Z}^d}$ such that $T = \sum_{m \in \mathbb{Z}^d} \lambda_m \, \varphi_m \otimes \varphi_{-m}$. Define

$$T_K := \sum_{m \in \mathbb{Z}^d_{\leq K}} \lambda_m \, \varphi_m \otimes \varphi_{-m}.$$

Clearly, $T_K \in \mathcal{T}_K$. Then, we have

$$
\begin{aligned}
r(T) - r(T_K) &= \mathop{\mathbb{E}}_{(v,w) \sim \mu} \left[ \|Tv - w\|_{L^2}^2 - \|T_K v - w\|_{L^2}^2 \right] \\
&= \mathop{\mathbb{E}}_{(v,w) \sim \mu} \left[ \|Tv\|_{L^2}^2 - \|T_K v\|_{L^2}^2 + 2 \left\langle (T_K - T)v, w \right\rangle \right] \\
&\leq \mathop{\mathbb{E}}_{(v,w) \sim \mu} \left[ \sum_{m \in \mathbb{Z}^d_{>K}} |\lambda_m|^2 |\langle \varphi_{-m}, v \rangle|^2 + 2 \sum_{m \in \mathbb{Z}^d_{>K}} \left| \lambda_m \langle \varphi_{-m}, v \rangle \langle \varphi_m, w \rangle \right| \right]
\end{aligned}
$$

The final equality uses the following facts. First, we have $\|Tv\|_{L^2}^2 = \left\| \sum_{m \in \mathbb{Z}^d} \lambda_m \langle \varphi_{-m}, v \rangle \varphi_m \right\|_{L^2}^2 = \sum_{m \in \mathbb{Z}^d} |\lambda_m|^2 |\langle \varphi_{-m}, v \rangle|^2$. Analogously, $\|T_K v\|_{L^2}^2 = \sum_{m \in \mathbb{Z}^d_{\leq K}} |\lambda_m|^2 |\langle \varphi_{-m}, v \rangle|^2$. As for the second term, we use

$$\left\langle (T_K - T)v, w \right\rangle = \left\langle \sum_{m \in \mathbb{Z}^d_{>K}} \lambda_m \langle \varphi_{-m}, v \rangle \varphi_m, w \right\rangle = \sum_{m \in \mathbb{Z}^d_{>K}} \lambda_m \langle \varphi_{-m}, v \rangle \langle \varphi_m, w \rangle.$$

Next, using the fact that $|\lambda_m| \leq C$ followed by Lemma D.3, the first term is

$$\sum_{m \in \mathbb{Z}^d_{>K}} |\lambda_m|^2 |\langle \varphi_{-m}, v \rangle|^2 \leq \frac{B^2 C^2}{K^{2s}}.$$

As for the second term, using $|\lambda_m| \leq C$ followed by Cauchy-Schwarz implies

$$2 \sum_{m \in \mathbb{Z}^d_{>K}} |\lambda_m \langle \varphi_{-m}, v \rangle \langle \varphi_m, w \rangle| \leq 2C \sqrt{\sum_{m \in \mathbb{Z}^d_{>K}} |\langle \varphi_{-m}, v \rangle|^2} \sqrt{\sum_{m \in \mathbb{Z}^d_{>K}} |\langle \varphi_m, w \rangle|^2} \leq \frac{2CB^2}{K^{2s}},$$

where the final inequality holds because of Lemma D.3. Since $T \in \mathcal{T}$ is arbitrary, we have shown that

$$\sup_{T \in \mathcal{T}} \inf_{T_K \in \mathcal{T}_K} |r(T) - r(T_K)| \leq \frac{B^2 C(C+2)}{K^{2s}} \leq \frac{B^2 (C+1)^2}{K^{2s}}.$$

**E.2   Upper bound on the discretization error** $2 \mathbb{E} \left[ \sup_{T \in \mathcal{T}_K} |\widehat{r}(T) - \widehat{r}_N(T)| \right]$

Fix $T \in \mathcal{T}_K$. Then, there exists $\{\lambda_m\}_{m \in \mathbb{Z}^d_{\leq K}}$ with $|\lambda_m| \leq C$ such that $T = \sum_{\mathbb{Z}^d_{\leq K}} \lambda_m \, \varphi_m \otimes \varphi_{-m}$. Then, recall that

$$\widehat{r}_N(T) := \frac{1}{n} \sum_{i=1}^n \sum_{m \in \mathbb{Z}^d_{\leq K}} \left| \lambda_m \, \mathrm{DFT}(v_i^N)(-m) - \mathrm{DFT}(w_i^N)(-m) \right|^2 + \frac{1}{n} \sum_{i=1}^n \sum_{m \in \mathbb{Z}^d_{>K}} |\langle \varphi_{-m}, w_i \rangle|^2.$$

Moreover, we also have

$$\widehat{r}(T) = \frac{1}{n} \sum_{i=1}^n \sum_{m \in \mathbb{Z}^d_{\leq K}} |\lambda_m \langle \varphi_{-m}, v_i \rangle - \langle \varphi_{-m}, w_i \rangle|^2 + \frac{1}{n} \sum_{i=1}^n \sum_{m \in \mathbb{Z}^d_{>K}} |\langle \varphi_{-m}, w_i \rangle|^2,$$

which yields

$$\widehat{r}_N(T) - \widehat{r}(T) = \frac{1}{n} \sum_{i=1}^{n} \sum_{m \in \mathbb{Z}_{\leq K}^d} \left( \left| \lambda_m \, \mathrm{DFT}(v_i^N)(-m) - \mathrm{DFT}(w_i^N)(-m) \right|^2 - \left| \lambda_m \left\langle \varphi_{-m}, v_i \right\rangle - \left\langle \varphi_{-m}, w_i \right\rangle \right|^2 \right).$$

Next, we define

$$\alpha_{im} = \mathrm{DFT}(v_i^N)(-m) - \left\langle \varphi_{-m}, v_i \right\rangle \quad \text{and} \quad \beta_{im} = \mathrm{DFT}(w_i^N)(-m) - \left\langle \varphi_{-m}, w_i \right\rangle.$$

We can then write

$$\begin{aligned}
& \left| \lambda_m \, \mathrm{DFT}(v_i^N)(-m) - \mathrm{DFT}(w_i^N)(-m) \right|^2 \\
&= \left| \lambda_m \left\langle \varphi_{-m}, v_i \right\rangle - \left\langle \varphi_{-m}, w_i \right\rangle + \lambda_m \, \alpha_{im} - \beta_{im} \right|^2 \\
&\leq \left| \lambda_m \left\langle \varphi_{-m}, v_i \right\rangle - \left\langle \varphi_{-m}, w_i \right\rangle \right|^2 + 2 \left| \lambda_m \left\langle \varphi_{-m}, v_i \right\rangle - \left\langle \varphi_{-m}, w_i \right\rangle \right| \left| \lambda_m \alpha_{im} - \beta_{im} \right| + \left| \lambda_m \alpha_{im} - \beta_{im} \right|^2.
\end{aligned}$$

Thus, we obtain

$$\begin{aligned}
|\widehat{r}_N(T) - \widehat{r}(T)| &\leq \frac{1}{n} \sum_{i=1}^{n} \sum_{m \in \mathbb{Z}_{\leq K}^d} \left( 2 \left| \lambda_m \left\langle \varphi_{-m}, v_i \right\rangle - \left\langle \varphi_{-m}, w_i \right\rangle \right| \left| \lambda_m \alpha_{im} - \beta_{im} \right| + \left| \lambda_m \alpha_{im} - \beta_{im} \right|^2 \right) \\
&\leq \max_{i \in [n]} \sum_{m \in \mathbb{Z}_{\leq K}^d} 2 \left( \left| \lambda_m \left\langle \varphi_{-m}, v_i \right\rangle \right| + \left| \left\langle \varphi_{-m}, w_i \right\rangle \right| \right) \left| \lambda_m \alpha_{im} - \beta_{im} \right| + \left| \lambda_m \alpha_{im} - \beta_{im} \right|^2.
\end{aligned}$$

Next, using Cauchy-Schwarz inequality, the first term of the summand can be bounded as

$$\begin{aligned}
& \sum_{m \in \mathbb{Z}_{\leq K}^d} \left| \lambda_m \left\langle \varphi_{-m}, v_i \right\rangle \right| \left| \lambda_m \alpha_{im} - \beta_{im} \right| \\
& \qquad \leq \sqrt{ \sum_{m \in \mathbb{Z}_{\leq K}^d} |\lambda_m|^2 (1 + |m|_\infty^{2s}) \left| \left\langle \varphi_{-m}, v_i \right\rangle \right|^2 } \sqrt{ \sum_{m \in \mathbb{Z}_{\leq K}^d} \frac{|\lambda_m \alpha_{im} - \beta_{im}|^2}{1 + |m|_\infty^{2s}} } \\
& \qquad \leq B\,C \sqrt{ \sum_{m \in \mathbb{Z}_{\leq K}^d} \frac{|\lambda_m \alpha_{im} - \beta_{im}|^2}{1 + |m|_\infty^{2s}} },
\end{aligned}$$

where the final inequality uses Lemma D.3 and the fact that $|\lambda_m| \leq C$. Similar arguments show that

$$\sum_{m \in \mathbb{Z}_{\leq K}^d} \left| \left\langle \varphi_{-m}, w_i \right\rangle \right| \left| (\lambda_m \alpha_{im} - \beta_{im}) \right| \leq B \sqrt{ \sum_{m \in \mathbb{Z}_{\leq K}^d} \frac{|\lambda_m \alpha_{im} - \beta_{im}|^2}{1 + |m|_\infty^{2s}} }.$$

Overall, we have shown that

$$|\widehat{r}_N(T) - \widehat{r}(T)| \leq \max_{i \in [n]} \left( 2B(C+1) \sqrt{ \sum_{m \in \mathbb{Z}_{\leq K}^d} \frac{|\lambda_m \alpha_{im} - \beta_{im}|^2}{1 + |m|_\infty^{2s}} } + \sum_{m \in \mathbb{Z}_{\leq K}^d} |\lambda_m \alpha_{im} - \beta_{im}|^2 \right).$$

Now, recall that Lemma D.6 implies

$$\max\{|\alpha_{im}|, |\beta_{im}|\} \leq \left| \sum_{\ell \in \mathbb{Z}^d \setminus \{\mathbf{0}\}} \left\langle \varphi_{-(\ell N + m)}, u \right\rangle \right|,$$

which subsequently yields

$$|\lambda_m \alpha_{im} - \beta_{im}|^2 \leq (C+1)^2 \left| \sum_{\ell \in \mathbb{Z}^d \backslash \{\mathbf{0}\}} \left\langle \varphi_{-(\ell N + m)}, u \right\rangle \right|^2.$$

Thus, we have

$$\sum_{m \in \mathbb{Z}^d_{\leq K}} |\lambda_m \alpha_{im} - \beta_{im}|^2$$

$$\leq (C+1)^2 \sum_{m \in \mathbb{Z}^d_{\leq K}} \left| \sum_{\ell \in \mathbb{Z}^d \backslash \{\mathbf{0}\}} \left\langle \varphi_{-(\ell N + m)}, u \right\rangle \right|^2$$

$$\leq (C+1)^2 \left( \sum_{m \in \mathbb{Z}^d_{\leq K}} \left( \sum_{\ell \in \mathbb{Z}^d \backslash \{\mathbf{0}\}} \frac{1}{1 + |m + \ell N|_{\infty}^{2s}} \right) \sum_{\ell \in \mathbb{Z}^d \backslash \{\mathbf{0}\}} (1 + |m + \ell N|_{\infty}^{2s}) \left| \left\langle \varphi_{-(\ell N + m)}, u \right\rangle \right|^2 \right),$$

where the final step follows from Cauchy-Schwarz inequality.

To upper bound the first sum within inner parenthesis, note that $|m + \ell N|_{\infty} \geq |\ell N|_{\infty} - |m|_{\infty} \geq N |\ell|_{\infty} - N/2 \geq N/2 \, |\ell|_{\infty}$. Here, we use the fact that $|m|_{\infty} \leq K \leq N/2$. So, we have

$$\sum_{\ell \in \mathbb{Z}^d \backslash \{\mathbf{0}\}} \frac{1}{1 + |m + \ell N|_{\infty}^{2s}} \leq \left( \frac{2}{N} \right)^{2s} \sum_{\ell \in \mathbb{Z}^d \backslash \{\mathbf{0}\}} \frac{1}{|\ell|_{\infty}^{2s}} \leq \frac{2^{2s} \pi^2 \, 3^{d-2}}{N^{2s}},$$

where the final inequality uses Lemma D.7. Next, note that

$$\sum_{m \in \mathbb{Z}^d_{\leq K}} \sum_{\ell \in \mathbb{Z}^d \backslash \{\mathbf{0}\}} (1 + |m + \ell N|_{\infty}^{2s}) \left| \left\langle \varphi_{-(\ell N + m)}, u \right\rangle \right|^2 \leq \sum_{k \in \mathbb{Z}^d} (1 + |k|_{\infty}^{2s}) \left| \left\langle \varphi_{-k}, u \right\rangle \right|^2 \leq B^2,$$

where the second inequality follows from Lemma D.3. The first inequality holds because for each $k \in \mathbb{Z}^d$, we have $|\{(m, \ell) : m + \ell N = k, m \in \mathbb{Z}^d_{\leq K} \text{ and } \ell \in \mathbb{Z}^d \backslash \{0\}\}| \leq 1$. That is, for each $k \in \mathbb{Z}^d$, there is only one possible pair $(m, \ell)$ such that $k = m + \ell N$. Suppose, for the sake of contradiction, there exists $k \in \mathbb{Z}^d$ such that two distinct pairs exist in the set, namely $(m_1, \ell_1)$ and $(m_2, \ell_2)$. Note that $m_1 + \ell_1 N - (m_2 + \ell_2 N) = k - k = 0$, which implies $(m_1 - m_2) = (\ell_2 - \ell_1)N$. Clearly, we cannot have $\ell_2 = \ell_1$, otherwise, we will have $m_2 = m_1$, contradicting the fact that there are two distinct pairs. So, we must have $\ell_2 \neq \ell_1$. That is, $|\ell_2 - \ell_1|_{\infty} \geq 1$, and thus $|m_1 - m_2|_{\infty} \geq N$. Moreover, $|m_1 - m_2|_{\infty} \leq |m_1|_{\infty} + |m_2|_{\infty} \leq 2K$, which implies that $2K \geq N$. This contradicts the fact that $K < N/2$. Therefore, overall, we have shown that

$$\sum_{m \in \mathbb{Z}^d_{\leq K}} |\lambda_m \alpha_{im} - \beta_{im}|^2 \leq \frac{2^{2s} \pi^2 \, 3^{d-2} \, B^2 (C+1)^2}{N^{2s}}.$$

Next, we have

$$\sqrt{\sum_{m \in \mathbb{Z}^d_{\leq K}} \frac{|\lambda_m \alpha_{im} - \beta_{im}|^2}{1 + |m|_{\infty}^{2s}}} \leq \sqrt{\sum_{m \in \mathbb{Z}^d_{\leq K}} |\lambda_m \alpha_{im} - \beta_{im}|^2} \leq \frac{2^s \pi \sqrt{3^{d-2}} \, B(C+1)}{N^s}.$$

Therefore, by combining everything, we have shown that

$$|\widehat{r}_N(T) - \widehat{r}(T)| \leq \frac{2^{s+1} B^2 (C+1)^2}{N^s} \pi \sqrt{3^{d-2}} + \frac{B^2 (C+1)^2 4^s}{N^{2s}} \pi^2 3^{d-2} \leq 2 \frac{2^{s+1} B^2 (C+1)^2}{N^s} \pi \sqrt{3^{d-2}}.$$

The final inequality holds when $N^s \geq 2^{s-1} \pi \sqrt{3^{d-2}}$, which is satisfied as long as $N \geq 6$. As $T \in \mathcal{T}_K$ is arbitrary, we have shown that the discretization error

$$2 \, \mathbb{E}\left[ \sup_{T \in \mathcal{T}_K} |\widehat{r}(T) - \widehat{r}_N(T)| \right] \leq \frac{2^{s+3} \pi \sqrt{3^{d-2}} B^2 (C+1)^2}{N^s} \leq \frac{2^{s+3} \sqrt{\pi^d} B^2 (C+1)^2}{N^s}.$$

### E.3 Upper Bound on the Statistical Error $3\,\mathbb{E}\left[\sup_{T\in\mathcal{T}_K}|r(T)-\widehat{r}(T)|\right]$

In fact, we will bound $\mathbb{E}\left[\sup_{T\in\mathcal{T}}|r(T)-\widehat{r}(T)|\right]$. This can be viewed as the limit of the statistical error as $K\to\infty$. To that end, let $\sigma_1,\dots,\sigma_n$ denote iid random variables such that $\sigma_i\sim\mathrm{Uniform}(\{-1,1\})$. Standard symmetrization arguments show that

$$
\begin{aligned}
\mathbb{E}\left[\sup_{T\in\mathcal{T}}|r(T)-\widehat{r}(T)|\right] &\le 2\,\mathbb{E}\left[\sup_{T\in\mathcal{T}}\left|\frac{1}{n}\sum_{i=1}^{n}\sigma_i\,\|Tv_i-w_i\|_{L^2}^2\right|\right]\\
&= 2\,\mathbb{E}\left[\sup_{|\lambda|_{\ell^\infty}\le C}\left|\frac{1}{n}\sum_{i=1}^{n}\sigma_i\sum_{m\in\mathbb{Z}^d}|\lambda_m\langle\varphi_{-m},v_i\rangle-\langle\varphi_{-m},w_i\rangle|^2\right|\right]
\end{aligned}
$$

Note that

$$
\begin{aligned}
&|\lambda_m\langle\varphi_{-m},v_i\rangle-\langle\varphi_{-m},w_i\rangle|^2\\
&= (\lambda_m\langle\varphi_{-m},v_i\rangle-\langle\varphi_{-m},w_i\rangle)\overline{(\lambda_m\langle\varphi_{-m},v_i\rangle-\langle\varphi_{-m},w_i\rangle)}\\
&= \lambda_m\overline{\lambda_m}\langle\varphi_{-m},v_i\rangle\overline{\langle\varphi_{-m},v_i\rangle}-\lambda_m\langle\varphi_{-m},v_i\rangle\overline{\langle\varphi_{-m},w_i\rangle}-\overline{\lambda_m}\langle\varphi_{-m},w_i\rangle\overline{\langle\varphi_{-m},v_i\rangle}+\langle\varphi_{-m},w_i\rangle\overline{\langle\varphi_{-m},w_i\rangle}\\
&= |\lambda_m|^2\,|\langle\varphi_{-m},v_i\rangle|^2-\left(\lambda_m\langle\varphi_{-m},v_i\rangle\overline{\langle\varphi_{-m},w_i\rangle}+\overline{\lambda_m}\langle\varphi_{-m},w_i\rangle\overline{\langle\varphi_{-m},v_i\rangle}\right)+|\langle\varphi_{-m},w_i\rangle|^2.
\end{aligned}
$$

The first and the last term above are real numbers, so the term in the parenthesis must also be a real number. Using triangle inequality, the term Rademacher sum above can be upper-bounded as

$$
\begin{aligned}
&\mathbb{E}\left[\sup_{|\lambda|_{\ell^\infty}\le C}\left|\frac{1}{n}\sum_{i=1}^{n}\sigma_i\sum_{m\in\mathbb{Z}^d}|\lambda_m\langle\varphi_{-m},v_i\rangle-\langle\varphi_{-m},w_i\rangle|^2\right|\right]\\
&\le \underbrace{\mathbb{E}\left[\sup_{|\lambda|_{\ell^\infty}\le C}\left|\frac{1}{n}\sum_{i=1}^{n}\sigma_i\sum_{m\in\mathbb{Z}^d}|\lambda_m|^2\,|\langle\varphi_{-m},v_i\rangle|^2\right|\right]}_{\text{(i)}}+\underbrace{\mathbb{E}\left[\sup_{|\lambda|_{\ell^\infty}\le C}\left|\frac{1}{n}\sum_{i=1}^{n}\sigma_i\sum_{m\in\mathbb{Z}^d}\lambda_m\langle\varphi_{-m},v_i\rangle\overline{\langle\varphi_{-m},w_i\rangle}\right|\right]}_{\text{(ii)}}\\
&+ \underbrace{\mathbb{E}\left[\sup_{|\lambda|_{\ell^\infty}\le C}\left|\frac{1}{n}\sum_{i=1}^{n}\sigma_i\sum_{m\in\mathbb{Z}^d}\overline{\lambda_m}\langle\varphi_{-m},w_i\rangle\overline{\langle\varphi_{-m},v_i\rangle}\right|\right]}_{\text{(iii)}}+\underbrace{\mathbb{E}\left[\left|\frac{1}{n}\sum_{i=1}^{n}\sigma_i\sum_{m\in\mathbb{Z}^d}|\langle\varphi_{-m},w_i\rangle|^2\right|\right]}_{\text{(iv)}}.
\end{aligned}
$$

Let us start with the term (iv) first. Swapping the sum over $m$ and $i$ and using triangle inequality yields

$$
\begin{aligned}
\text{(iv)} = \mathbb{E}\left[\left|\frac{1}{n}\sum_{i=1}^{n}\sigma_i\sum_{m\in\mathbb{Z}^d}|\langle\varphi_{-m},w_i\rangle|^2\right|\right] &\le \sum_{m\in\mathbb{Z}^d}\frac{1}{n}\mathbb{E}\left[\left|\sum_{i=1}^{n}\sigma_i\,|\langle\varphi_{-m},w_i\rangle|^2\right|\right]\\
&\le \sum_{m\in\mathbb{Z}^d}\frac{1}{n}\left(\sum_{i=1}^{n}|\langle\varphi_{-m},w_i\rangle|^4\right)^{1/2},
\end{aligned}
$$

where the final step follows from Khintchine's inequality. Note that swapping the sums is justified because both sums converge absolutely.

For the term (iii), swapping the sum over $m$ and $i$ and using the fact that $|\lambda_m| \le C$ yields

$$(\mathrm{iii}) = \mathbb{E}\left[\sup_{|\lambda|_{\ell^\infty} \le C} \left|\frac{1}{n}\sum_{i=1}^{n}\sigma_i\sum_{m\in\mathbb{Z}^d}\overline{\lambda_m}\left\langle\varphi_{-m},w_i\right\rangle\overline{\left\langle\varphi_{-m},v_i\right\rangle}\right|\right]$$

$$= \mathbb{E}\left[\sup_{|\lambda|_{\ell^\infty} \le C} \left|\frac{1}{n}\sum_{m\in\mathbb{Z}^d}\overline{\lambda_m}\sum_{i=1}^{n}\sigma_i\left\langle\varphi_{-m},w_i\right\rangle\overline{\left\langle\varphi_{-m},v_i\right\rangle}\right|\right]$$

$$\le C\,\mathbb{E}\left[\sum_{m\in\mathbb{Z}^d}\left|\frac{1}{n}\sum_{i=1}^{n}\sigma_i\left\langle\varphi_{-m},w_i\right\rangle\overline{\left\langle\varphi_{-m},v_i\right\rangle}\right|\right]$$

$$\le C\sum_{m\in\mathbb{Z}^d}\frac{1}{n}\left(\sum_{i=1}^{n}|\left\langle\varphi_{-m},w_i\right\rangle\overline{\left\langle\varphi_{-m},v_i\right\rangle}|^2\right)^{1/2},$$

where the final step uses Khintchine's inequality. Since $|\lambda_m| \le C$, we can use the same arguments to show that

$$(\mathrm{ii}) \le C\sum_{m\in\mathbb{Z}^d}\frac{1}{n}\left(\sum_{i=1}^{n}|\left\langle\varphi_{-m},v_i\right\rangle\overline{\left\langle\varphi_{-m},w_i\right\rangle}|^2\right)^{1/2},$$

and

$$(\mathrm{i}) \le C^2\sum_{m\in\mathbb{Z}^d}\frac{1}{n}\left(\sum_{i=1}^{n}|\left\langle\varphi_{-m},v_i\right\rangle|^4\right)^{1/2}.$$

Next, note that we can bound $|\left\langle\varphi_0,u\right\rangle| \le B$ for all $\|u\|_{\mathcal{H}^s} \le B$. Moreover, Lemma D.1 implies that $|\left\langle\varphi_{-m},u\right\rangle| \le \frac{B}{(2\pi)^s|m|_\infty^s}$ for all $m \ne \mathbf{0}$. Thus, we obtain the bound

$$(\mathrm{i}) \le \frac{B^2C^2}{\sqrt{n}} + C^2\sum_{m\in\mathbb{Z}^d\setminus\{\mathbf{0}\}}\frac{1}{n}\left(\sum_{i=1}^{n}\frac{B^4}{(2\pi)^{4s}}\frac{1}{|m|_\infty^{4s}}\right)^{1/2}$$

$$\le B^2C^2\frac{1}{\sqrt{n}} + \frac{B^2C^2}{(2\pi)^{2s}}\frac{1}{\sqrt{n}}\sum_{m\in\mathbb{Z}^d\setminus\{\mathbf{0}\}}\frac{1}{|m|_\infty^{2s}}$$

$$\le B^2C^2\frac{1}{\sqrt{n}} + \frac{B^2C^2\pi^2 3^{d-2}}{(2\pi)^{2s}}\frac{1}{\sqrt{n}},$$

where the final inequality uses Lemma D.7. Similar calculations can be done to show that

$$(\mathrm{ii}),(\mathrm{iii}) \le B^2C\frac{1}{\sqrt{n}} + \frac{B^2C\pi^2 3^{d-2}}{(2\pi)^{2s}}\frac{1}{\sqrt{n}} \quad\text{and}\quad (\mathrm{iv}) \le B^2\frac{1}{\sqrt{n}} + \frac{B^2\pi^2 3^{d-2}}{(2\pi)^{2s}}\frac{1}{\sqrt{n}}.$$

Thus, we have overall shown that

$$\mathbb{E}\left[\sup_{T\in\mathcal{T}}|r(T)-\widehat{r}(T)|\right] \le 2\left((\mathrm{i})+(\mathrm{ii})+(\mathrm{iii})+(\mathrm{iv})\right)$$

$$\le 2(B^2C^2+2B^2C+B^2)\left(1+\frac{\pi^2 3^{d-2}}{(2\pi)^{2s}}\right)\frac{1}{\sqrt{n}}$$

$$= \frac{2B^2(C+1)^2}{\sqrt{n}}\left(1+\frac{\pi^2 3^{d-2}}{(2\pi)^{2s}}\right)$$

$$\le \frac{5}{2}\frac{B^2(C+1)^2}{\sqrt{n}}$$

where we use the fact that

$$\frac{\pi^2 3^{d-2}}{(2\pi)^{2s}} \le \frac{1}{2^{2s}}\frac{\pi^d}{\pi^{2s}} \le \frac{1}{2^{2s}} \le \frac{1}{4}$$

as $2s > d$ and $s \geq 1$. Therefore, the overall statistical error is

$$3\,\mathbb{E}\left[\sup_{T \in \mathcal{T}} |r(T) - \widehat{r}(T)|\right] \leq \frac{8B^2(C+1)^2}{\sqrt{n}}.$$

## F  Proof of Lower Bound (Theorem 3.3)

*Proof.* To define a difficult distribution for the learner, we need some notations. Let

$$\psi_0 = \varphi_0 \quad \text{and } \psi_m = \frac{1}{\sqrt{2}}\left(\varphi_{-m} + \varphi_m\right) \quad \text{for } m \in \mathbb{Z}^d \backslash \{\mathbf{0}\}.$$

Note that $\psi_m : \mathbb{T}^d \to \mathbb{R}$ is a *real-valued* function such that $\|\psi_m\|_{L^2} = 1$. We work with $\psi_m$'s to ensure that the distribution is only supported over real-valued functions. For any $\{\lambda_k\}_{k \in \mathbb{Z}^d}$ such that $\lambda_k = \lambda_{-k} \in \mathbb{R}$, the operator $T = \sum_{m \in \mathbb{Z}^d} \lambda_m \varphi_m \otimes \varphi_{-m}$ satisfies

$$T\psi_m = \frac{1}{\sqrt{2}}(\lambda_m\,\varphi_m + \lambda_{-m}\varphi_{-m}) = \frac{\lambda_m}{\sqrt{2}}\left(\varphi_{-m} + \varphi_m\right) = \lambda_m\psi_m \quad \forall m \in \mathbb{Z}^d\backslash\{\mathbf{0}\}.$$

Clearly, $T\psi_0 = \lambda_0\psi_0$. Next, let us define a sequence $\{\gamma_m\}_{m \in \mathbb{Z}^d}$ such that

$$\gamma_0 = \frac{B}{\sqrt{s+1}} \quad \text{and} \quad \gamma_m = \frac{B}{\sqrt{s+1}\,|m|_\infty^s} \quad \forall m \in \mathbb{Z}^d\backslash\{\mathbf{0}\}.$$

Finally, define a set

$$\mathcal{J} = \{m \in \mathbb{Z}^d \; : \; m_1 \in \mathbb{N} \text{ and } m_j = 0 \quad \forall j \neq 1\}.$$

For any $M, N \in \mathbb{N}$, define $\mathcal{J}_M^N = \{m \in \mathcal{J} \; : \; m_1 \not\equiv 0 \pmod{N} \text{ and } m_1 \leq M\}$. Let $r \in \mathbb{Z}^d$ such that $r \in \mathcal{J}$ and $r_1 = 1$. That is, $r = (1, 0, 0, \dots, 0)$. For any $q \in \mathbb{Z}$, we write $qr = (q, 0, 0, \dots, 0)$.

We now describe a difficult distribution for the learner. To that end, first draw a $\xi := \{\xi_m\}_{m \in \mathbb{Z}^d}$ such that $\xi_m = \xi_{-m}$ is drawn from Uniform($\{-1, 1\}$). Then, given such $\xi$, let $\mu_\xi$ be any joint distribution on $\mathcal{V} \times \mathcal{W}$ such that its marginal on $\mathcal{V}$ assigns $1/3$ mass uniformly on $\{\gamma_m\psi_m \; : \; m \in \mathcal{J}_M^N\}$, $1/3$ mass on $\gamma_0\psi_0$, and the remaining $1/3$ mass on $\gamma_{(K+j)r}\,\psi_{(K+j)r}$ for either $j = 1$ or $j = 2$ ensuring that $K + j \not\equiv 0 \pmod{N}$. Moreover, given a $v = \gamma_k\psi_k$ drawn from the marginal of $\mu_\xi$, assign $w \mid v$ to be $\xi_k\gamma_k\psi_k$ if $k \neq 0$. On the other hand, if $k = 0$, then $w \mid v$ is $\xi_{Nr}\,\gamma_{Nr}\,\psi_{Nr}$.

This is a valid distribution as

$$\begin{aligned}
\|v\|_{\mathcal{H}^s}^2 &= \sum_{k \in \mathbb{N}_0^d \,:\, |k|_\infty \leq s} \left\|\partial^k v\right\|_{L^2}^2 = \sum_{k \in \mathbb{N}_0^d \,:\, |k|_\infty \leq s} (m_1^{k_1}\gamma_m)^2 \mathbb{1}[k_j = 0 \text{ for all } j \neq 1] \\
&= \gamma_m^2 \sum_{k_1=0}^s |m|_\infty^{2k_1} \\
&\leq (s+1)\gamma_m^2|m|_\infty^{2s} \\
&\leq B^2
\end{aligned}$$

Similar arguments show that $\|w\|_{\mathcal{H}^s}^2 \leq B^2$.

Next, we establish that

$$\mathbb{E}_\xi\left[\mathcal{E}_n(\widehat{T}_K^N, \mathcal{T}, \mu_\xi)\right] \geq \frac{B^2}{3(s+1)}\left(\frac{1}{8n} + \frac{2}{(K+2)^{2s}} + \frac{1}{N^{2s}}\right).$$

Since the lower bound above holds in expectation, we can use the probabilistic method to argue that there must exist a sequence $\xi^\star$ such that $\mathcal{E}_n(\widehat{T}_K^N, \mathcal{T}, \mu_{\xi^\star}) \geq \frac{B^2}{3(s+1)}\left(\frac{1}{8n} + \frac{2}{(K+2)^{2s}} + \frac{1}{N^{2s}}\right)$.

We now proceed with the proof of the claimed lowerbound. Let $\widehat{T}_K^N$ denote the estimator produced by the algorithm. Then, there exists $\{\widehat{\lambda}_m\}_{m \in \mathbb{Z}_{\leq K}^d}$ such that

$$\widehat{T}_K^N = \sum_{m \in \mathbb{Z}_{\leq K}^d} \widehat{\lambda}_m \, \varphi_m \otimes \varphi_{-m}.$$

For convenience, we will extend the sum to the entire $\mathbb{Z}^d$ and write $\widehat{T}_K^N = \sum_{m \in \mathbb{Z}^d} \widehat{\lambda}_m \, \varphi_m \otimes \varphi_{-m}$, where $\widehat{\lambda}_m = 0$ for all $m \in \mathbb{Z}_{> K}^d$.

Given a $\xi$, we now lowerbound the expected loss of $\widehat{T}_K^N$ on $\mu_\xi$. Using the definition of the distribution $\mu_\xi$, we have

$$\mathop{\mathbb{E}}_{(v,w) \sim \mu_\xi} \left[ \|\widehat{T}_K^N v - w\|_{L^2}^2 \right]$$

$$= \frac{1}{3} \frac{1}{|\mathcal{J}_M^N|} \sum_{m \in \mathcal{J}_M^N} \left( \widehat{\lambda}_m - \xi_m \right)^2 \gamma_m^2 + \frac{1}{3} \left\| \widehat{\lambda}_0 \gamma_0 \psi_0 - \xi_{Nr} \, \gamma_{Nr} \, \psi_{Nr} \right\|_{L^2}^2 + \frac{1}{3} \left\| 0 \, \psi_{(K+j)r} - \gamma_{(K+j)r} \, \psi_{(K+j)r} \right\|_{L^2}^2$$

$$\geq \frac{1}{3|\mathcal{J}_M^N|} \sum_{m \in \mathcal{J}_M^N} \gamma_m^2 \, \mathbb{1}[\widehat{\lambda}_m \xi_m \leq 0] + \frac{\widehat{\lambda}_0^2 \gamma_0^2 + \gamma_{Nr}^2}{3} + \frac{\gamma_{(K+j)r}^2}{3}$$

$$\geq \frac{\gamma_r^2}{3|\mathcal{J}_M^N|} \, \mathbb{1}[\widehat{\lambda}_r \xi_r \leq 0] + \frac{\widehat{\lambda}_0^2 \gamma_0^2 + \gamma_{Nr}^2}{3} + \frac{\gamma_{(K+2)r}^2}{3}.$$

Here, the first inequality use the fact that $(\widehat{\lambda}_m - \xi_m)^2 \geq 1$ whenever $\widehat{\lambda}_m \xi_m \leq 0$ and $\langle e_0, e_{Nr} \rangle_{L^2} = 0$. The second inequality uses the fact that $r \in \mathcal{J}_M^N$ as long as $M, N > 1$ and that $\gamma_{(K+j)r}^2 \geq \gamma_{(K+2)r}^2$ for $j \in \{1, 2\}$.

Next, we establish the upper bound on the loss of the best-fixed operator. Given $\xi$, define an operator

$$T_\xi = \sum_{m \in \mathbb{Z}_{>0}^d} \xi_m \, \varphi_m \otimes \varphi_{-m}.$$

Clearly,

$$\inf_{T \in \mathcal{T}} \mathop{\mathbb{E}}_{(v,w) \sim \mu_\xi} \left[ \|Tv - w\|_{L^2}^2 \right]$$

$$\leq \mathop{\mathbb{E}}_{(v,w) \sim \mu_\xi} \left[ \|T_\xi v - w\|_{L^2}^2 \right]$$

$$= \mathbb{E}\left[ \|T_\xi v - w\|_{L^2}^2 \,\big|\, v = \gamma_0 \psi_0 \right] \mathbb{P}[v = \gamma_0 \psi_0] + \mathbb{E}\left[ \|T_\xi v - w\|_{L^2}^2 \,\big|\, v \neq \gamma_0 \psi_0 \right] \mathbb{P}[v \neq \gamma_0 \psi_0]$$

$$\leq \|0 - \xi_{Nr} \, \gamma_{Nr} \, \psi_{Nr}\|_{L^2}^2 \, \frac{1}{3}$$

$$\leq \frac{\gamma_{Nr}^2}{3},$$

where we use the fact that $T_\xi v = 0$ whenever $v = \gamma_0 e_0$ and $T_\xi v = w$ otherwise. Overall, we have shown that

$$\mathop{\mathbb{E}}_{(v,w) \sim \mu_\xi} \left[ \|\widehat{T}_K^N v - w\|_{L^2}^2 \right] - \inf_{T \in \mathcal{T}} \mathop{\mathbb{E}}_{(v,w) \sim \mu_\xi} \left[ \|Tv - w\|_{L^2}^2 \right]$$

$$\geq \frac{\gamma_r^2}{3|\mathcal{J}_M^N|} \, \mathbb{1}[\widehat{\lambda}_r \xi_r \leq 0] + \frac{\widehat{\lambda}_0^2 \gamma_0^2 + \gamma_{Nr}^2}{3} + \frac{\gamma_t^2}{3} - \frac{\gamma_{Nr}^2}{3}$$

$$\geq \frac{1}{3(s+1)} \left( \frac{\mathbb{1}[\widehat{\lambda}_r \xi_r \leq 0]}{|\mathcal{J}_M^N|} + \widehat{\lambda}_0^2 + \frac{B^2}{(K+2)^{2s}} \right),$$

where the final inequality holds because $\gamma_0 = \gamma_r = \frac{B}{\sqrt{s+1}}$ and $\gamma_{(K+2)r} = \frac{B}{\sqrt{s+1}(K+2)^{2s}}$.

Next, we establish lowerbound of $\widehat{\lambda}_0^2$. To that end, let $S_n = \{(v_i, w_i)\}_{i=1}^n$ denote the $n$ samples accessible to the learner over the uniform grid of size $N$. Recall our notation $v_i^N := \{v_i(x) : x \in \mathrm{G}\}$ and $w_i^N := \{w_i(x) :$

$x \in$ G} for discretized samples. Take a sample $(v_i, w_i) \sim \mu_\xi$. Then, we must have $v_i = \gamma_k \psi_k$ for some $k \in \mathbb{Z}^d$. Consider the case that $k \neq 0$. Then, by definition of the distribution $\mu_\xi$, it must be the case that $k \not\equiv 0 \pmod{} N$. Then, Lemma D.5 implies that

$$\mathrm{DFT}(v_i^N)(-0) = \frac{1}{N^d} \sum_{x \in \mathrm{G}} \gamma_k \, \psi_k(x) \, e^{-2\pi \mathrm{i} \langle x, 0 \rangle} = \frac{\gamma_k}{\sqrt{2} N^d} \left( \sum_{x \in \mathrm{G}} e^{-2\pi \mathrm{i} \langle k, x \rangle} + \sum_{x \in \mathrm{G}} e^{2\pi \mathrm{i} \langle k, x \rangle} \right) = 0.$$

On the other hand, if $v_i = \gamma_0 \psi_0$, then we have

$$\mathrm{DFT}(v_i^N)(-0) = \frac{1}{N^d} \sum_{x \in \mathrm{G}} \gamma_0 \psi_0(x) = \frac{\gamma_0}{N^d} \sum_{x \in \mathrm{G}} 1 = \gamma_0.$$

Additionally, when $v_i = \gamma_0 \psi_0$, we have $w_i = \gamma_{Nr} \psi_{Nr}$. In this case, Lemma D.5 implies that

$$\mathrm{DFT}(w_i^N)(-0) = \frac{\gamma_{Nr}}{N^d} \sum_{x \in \mathrm{G}} \psi_{Nr}(x) = \frac{\gamma_{Nr}}{\sqrt{2} N^d} \left( \sum_{x \in \mathrm{G}} e^{-2\pi \mathrm{i} \langle Nr, x \rangle} + \sum_{x \in \mathrm{G}} e^{2\pi \mathrm{i} \langle Nr, x \rangle} \right) = \frac{\gamma_{Nr}}{\sqrt{2}} 2 = \sqrt{2} \gamma_{Nr}.$$

Using these facts, we can write the empirical least-square loss as

$$\frac{1}{n} \sum_{i=1}^n \sum_{m \in \mathbb{Z}_{\leq K}^d} \left| \lambda_m \, \mathrm{DFT}(v_i^N)(-m) - \mathrm{DFT}(w_i^N)(-m) \right|^2$$

$$= \frac{|\lambda_0 - \sqrt{2} \gamma_{Nr}|^2}{n} \sum_{i=1}^n \mathbb{1}[v_i = \gamma_0 \psi_0] \; + \frac{1}{n} \sum_{i=1}^n \mathbb{1}[v_i \neq \gamma_0 \psi_0] \left| \mathrm{DFT}(w_i^N)(-m) \right|^2$$

$$+ \frac{1}{n} \sum_{i=1}^n \sum_{m \in \mathbb{Z}_{\leq K}^d \setminus \{\mathbf{0}\}} \left| \lambda_m \, \mathrm{DFT}(v_i^N)(-m) - \mathrm{DFT}(w_i^N)(-m) \right|^2$$

Thus, the least squares estimator for $\lambda_0$ must be $\widehat{\lambda}_0 = \sqrt{2} \gamma_{Nr}$. That is,

$$\widehat{\lambda}_0^2 = 2 \gamma_{Nr}^2 = \frac{2B^2}{(s+1)|Nr|_\infty^s} = \frac{2B^2}{(s+1)N^{2s}}.$$

Note that this choice of $\widehat{\lambda}_0$ is valid as $\widehat{\lambda}_0 \leq 1$. Thus, so far, we have shown that

$$\mathop{\mathbb{E}}_{(v,w) \sim \mu_\xi} \left[ \| \widehat{T}_K^N v - w \|_{L^2}^2 \right] - \mathop{\mathbb{E}}_{(v,w) \sim \mu_\xi} \left[ \| T_\xi v - w \|_{L^2}^2 \| \right] \geq \frac{B^2}{3(s+1)} \left( \frac{\mathbb{1}[\widehat{\lambda}_r \xi_r \leq 0]}{|\mathcal{J}_M^N|} + \frac{2}{N^{2s}} + \frac{1}{(K+2)^{2s}} \right)$$

Our proof will be complete upon establishing that

$$\frac{1}{|\mathcal{J}_M^N|} \mathop{\mathbb{E}}_\xi \left[ \mathop{\mathbb{E}}_{S_n \sim \mu_\xi} \left[ \mathbb{1}[\widehat{\lambda}_r \xi_r \leq 0] \right] \right] \geq \frac{1}{8n}$$

for an appropriate choice of $M$. To that end, let $\mu_\xi^\mathcal{V}$ be the marginal of $\mu_\xi$ on $\mathcal{V}$ and $S_n^\mathcal{V} \in \mathcal{V}^n$ denote the restriction of samples $S_n \in (\mathcal{V} \times \mathcal{W})^n$ to its first arguments. Then, we can change the order of expectations to write

$$\mathop{\mathbb{E}}_\xi \left[ \mathop{\mathbb{E}}_{S_n \sim \mu_\xi} \left[ \mathbb{1}[\widehat{\lambda}_r \xi_r \leq 0] \right] \right] = \mathop{\mathbb{E}}_{S_n^\mathcal{V} \sim \mu_n^\mathcal{V}} \left[ \mathop{\mathbb{E}}_\xi \left[ \mathbb{1}[\widehat{\lambda}_r \xi_r \leq 0] \right] \right] \geq \frac{1}{2} \mathbb{P}[\gamma_r \psi_r \notin S_n^\mathcal{V}]$$

To understand why the final inequality holds, observe that when the event $\gamma_r \psi_r \notin S_n^\mathcal{V}$ occurs, the learner has no information about $\xi_r$. This implies that $\xi_r$ and $\widehat{\lambda}_r$ are independent. Consequently, given that $\gamma_r \psi_r \notin S_n^\mathcal{V}$, the event $\widehat{\lambda}_r \xi_r \leq 0$ has a probability of at least $1/2$ since $\xi_r$ is sampled uniformly from $\{-1, +1\}$.

Next, it remains to pick $M$ such that

$$\frac{\mathbb{P}[\gamma_r \psi_r \notin S_n^\mathcal{V}]}{|\mathcal{J}_M^N|} \geq \frac{1}{4n}.$$

To get this, we choose $M = 2n$. It is easy to verify that $|\mathcal{J}_M^N| \geq n$ whenever $N > 1$. This is true because no more than half of integers in $\{1, 2, \ldots, 2n\}$ are divisible by $N$. Thus, we have

$$\mathbb{P}[\gamma_r \psi_r \notin S_n^{\mathcal{V}}] = \left(1 - \frac{1}{3|\mathcal{J}_M^N|}\right)^n \geq \left(1 - \frac{1}{3n}\right)^n \geq \frac{1}{2}$$

for any $n \geq 1$. Noting that $|\mathcal{J}_M^N| \leq 2n$ completes our proof. $\qquad\square$

## G   Additional Experiments

In this section, we present additional experiments with y-axis on log scale to illustrate the decay rate. Each plot includes the fitted slope $s$ and the corresponding smoothness parameter $\gamma$. The data generation, training, and evaluation setup follow Section 4. For each input function $v$ sampled at smoothness $\gamma$, the noise term $\varepsilon$ is sampled at smoothness $1.5 \cdot \gamma$. Figures 4, 5, and 6 plot the statistical, truncation, and discretization errors, respectively. In each panel the reported value $m$ is the slope of a line fitted to $\log(\text{error})$ versus $\log n$, $\log K$, or $\log N$, as appropriate.

The statistical error decays faster than our predicted $1/\sqrt{n}$ rate, with slopes close to or better than $-1.0$, even better than our lower bound. This may be due to the Gaussian distribution used in experiments, rather than worst-case inputs. Additionally, the error appears to vary slightly with $\gamma$, suggesting that smoother inputs might allow sharper statistical rates, at least for well-behaved distributions.

For truncation error, the observed rates are generally faster than predicted for $\gamma = 1.5, 2$, and slightly slower (but close) for $\gamma = 3$. Recall that our theory predicts $m$ to be $2\gamma$.

Interestingly, the discretization error remains nearly constant with a slope around $-1.1$ across all values of $\gamma$, whereas the theory predicts the estimated slope to be $\gamma$. This discrepancy may stem from the experimental setup: our theory considers the test resolution $N_2 \to \infty$, while the experiments fix a finite $N_2 = 512$. A more detailed analysis under finite training and testing resolution could help clarify this behavior.

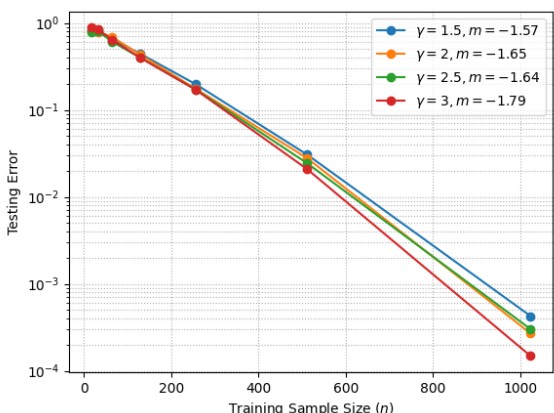

Figure 4: Statistical error decay across sample sizes for different smoothness values $\gamma$.

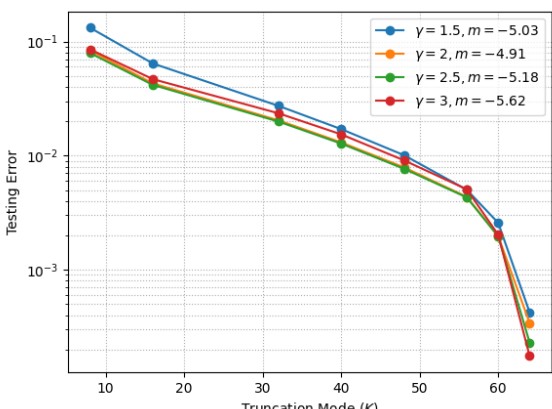

Figure 5: Truncation error plotted against truncation mode for various values of $\gamma$.

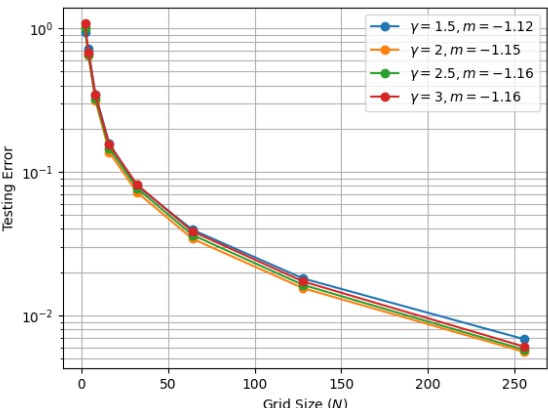

Figure 6: Discretization error as a function of grid resolution for various smoothness levels $\gamma$.

