# OpenReview forum: "Controlling Statistical, Discretization, and Truncation Errors in Learning Fourier Linear Operators"
_TMLR — Accepted by TMLR_

### Review · Reviewer_WNfE · 2025-06-13

**Summary Of Contributions:**

Fourier Neural Operators (FNOs) are widely used for operator learning, but their theoretical understanding remains limited. This paper focuses on the linear transform core of FNOs—analogous to the affine layer Wx + b in MLPs—and analyzes its statistical learning properties. The authors study the excess risk and derive bounds for three key sources of error: statistical error (from finite sample size), truncation error (from discarding high-frequency modes), and discretization error (from evaluating functions on a finite grid)

**Audience:**

Yes

**Broader Impact Concerns:**

I do not foresee any ethical risks associated with this work.

**Claims And Evidence:**

Yes

**Requested Changes:**

- The claim that “some of our proof techniques are inspired by their work Lanthaler et al. (2024)” is too vague. The manuscript should specify exactly which lemmas or arguments are reused or adapted.

- Reproduce the error plots on log–log scales and report the slope of fitted lines. See Lanthaler et al. (2024) for examples.

- Plot the curves for a few different smoothness parameter s.

**Strengths And Weaknesses:**

Strength:
- The paper is clearly written, self-contained, and highlights both connections to and differences from prior work. While I only reviewed the appendices superficially due to their length and time constraints, the main technical development appears sound.

- To the best of my knowledge, this is the first attempt of statistical learning properties in FNO, even if in a simplified setting. As such, it may serve as a useful foundation for future work. The authors are upfront about the limitations of their analysis, including the restriction to linear operators, periodic domains, and scalar-valued functions.

Weakness:
- The numerical experiments are limited in value. They show monotonic error decay but do not demonstrate the convergence rates predicted by theory.

- A bound that depends on N and K is valuable, as it helps clarify whether increasing grid resolution or spectral bandwidth is more effective under a fixed computational budget. The paper would be more impactful with a discussion of the computational trade-offs between N and K, along with simple experiments to validate and illustrate this comparison.

---

> ### Author Response · Authors · 2025-07-12
> **Authors Response to Reviewer WNfe**
>
> We thank the reviewer for the careful reading and thoughtful suggestions. We appreciate the recognition of the technical contributions and the clarity of the presentation.
>
> 1. We have revised the manuscript to be more specific about the fact that lemmas D.5 and D.6 are derived primarily from [1].
>
> 2.  We have included a section of additional experiments in Appendix where the curves are presented in log scale.
>
> 3. As suggested, we now plot curves for multiple values of the smoothness parameter $\gamma$ in each experiment.
>
> [1] Samuel Lanthaler, Andrew M Stuart, and Margaret Trautner. Discretization error of fourier neural operators.
> arXiv preprint arXiv:2405.02221, 2024.

---

### Review · Reviewer_LPXe · 2025-06-27

**Summary Of Contributions:**

This paper is a theoretical analysis of learning linear operators between function spaces. The authors divide the error into statistical, discretisation and truncation errors, and analyse them separately, obtaining upper and lower bounds (not quite matching).

**Audience:**

Yes

**Broader Impact Concerns:**

No broader impact concerns.

**Claims And Evidence:**

Yes

**Requested Changes:**

Page 3, "Our work aligns with the historical development of neural networks theory where the statistical properties of the linear core $x\mapsto Wx+b$ (a linear regression problem) were fully understood before studying deep neural networks." This is quite a bold yet simplifying statement. There are still many unsolved problems in linear regression, many unrelated to the analysis of deep neural networks, and moreover, much of the analysis of deep learning is not inspired by linear regression. Of course I see the point that the authors are trying to make but I would change or remove this sentence.

Page 4, Section 2.2: The complex conjugate notation was already introduced in Section 2.1. Also I'm a bit confused here - the codomain of $L^2(\mathbb{T}^d,\mathbb{R})$ is specified to be the space of real numbers, so why are you considering complex conjugates at all? Or should you change the codomain to $\mathbb{C}$? I think you should - in fact, the Fourier basis functions $\varphi_m$ map to $\mathbb{C}$ by definition. You should also change the codomain of $\varphi_m$ in Section 2.1 to $\mathbb{C}$ as well. Please have a look at this throughout the paper.

**Minor Comments**

Page 2, Section 1.1, two lines below the displayed equation: a integral kernel -> an integral kernel

Page 2, Section 1.2: "When $|\Lambda_\beta(m)|_{\ell_1}$" technically this should be something like the $\ell_1$ norm of a sequence indexed by $m\in\mathbb{Z}^d$. As currently written, you are taking the $\ell_1$ norm of a single number for a single $m$.

Page 5, section 2.3: In the definition of Sobolev spaces, I think it would be better to replace the "s.t." with "for all". Also, in the last paragraph of section 2.3, you say that sometimes one takes the 2-norm of k? I've never seen this, I've only ever seen the 1-norm of k in the definition of Sobolev spaces.

Page 5, section 3, second paragraph: we will also $\lambda$ -> we will also use $\lambda$.

Page 5, equation (1): I think it would be better to write $(\cdot)$ on the right-hand side too, or even $(u)$ on both sides, since you say "for every $u$" afterwards.

Page 7, Section 3.2, paragraph beginning with "To see why", second line: $v_1$ should be $v_i$.

**Strengths And Weaknesses:**

**Strengths**

I do not have much background in working in operator learning, but I am convinced that the problem that the authors are trying to tackle is an important and interesting question. The authors do a great job of introducing the problem, and pitching it against existing works. Although I didn't have time to go through the proofs in detail, the results are very believable, and are in line with standard results.

The lower bound does not quite match the upper bound, but I do not see this as a weakness. The authors didn't have to report the lower bound, and I think the fact that they put in the effort to derive it and reported it should be commended.

I also love how thorough the preliminary section is. It really makes the paper self-contained and easy to follow. In general, the whole paper is written in a way that is a pleasure to read. Thank you very much.

**Weaknesses**

See "Requested Changes" section.

---

> ### Author Response · Authors · 2025-07-12
> **Authors Response to Reviewer LPXe**
>
> We thank the reviewer for the thoughtful feedback, especially for the recognition of the clarity of our preliminary section. We address the specific comments below.
>
> 1. We recognize the reviewer’s point and agree that the original sentence may have been overly simplified. We have removed that particular sentence in revised manuscript.
>
> 2. Thank you for flagging inconsistency in complex conjugates and codomain. As suggested, we now explicitly state that the Fourier basis functions $\varphi_m$ map to $\mathbb{C}$, even though the ambient space $L^2(\mathbb{T}^d, \mathbb{R})$ contains real-valued functions. We have revised Section 2.1 accordingly and ensured consistency throughout the paper.
>
> 3. We have addressed all minor comments.

---

### Review · Reviewer_Fggu · 2025-07-03

**Summary Of Contributions:**

This paper develops a theoretical framework for understanding operator learning through the lens of a simplified linear within FNO. The authors identify and analyze three key sources of error—statistical, truncation, and discretization—and provide precise upper and lower bounds for each. Their results offer new insights into multiresolution generalization and lay groundwork for a deeper statistical understanding of neural operator architectures.

**Audience:**

Yes

**Broader Impact Concerns:**

Like previously said, the focus on linearality core of FNO and assumption on uniform grid significantly limits the paper for broader impact. How does this prevent the community from apply to a broader use cases will significantly help reader to understand the contribution of the work.

**Claims And Evidence:**

Yes

**Requested Changes:**

1. Explicitly explain in Section 4 how the empirical results relate back to the theoretical error bounds (e.g., do they validate the predicted convergence rates? Do they expose gaps in the bounds?). Please summarize the key takeaway and its implications for the broader theoretical message

2. Add a discussion (possibly in Section 5) that more explicitly outlines how the linear results might extend to nonlinear operators and multi-layer FNOs, and what technical obstacles exist

3. Acknowledge the restriction to uniform grids more explicitly and suggest potential strategies for handling non-uniform or unstructured data (e.g., quadrature, random sampling, interpolation)

**Strengths And Weaknesses:**

Weakness:
1. I don't see how the section 4 connects to previous sections. What is the conclusion of section 4 and how does that help to enhance the main message of the paper?
2. The information density in Figure 1/2/3 is relatively low. The conclusion draw from the three figures seems well known. Considering plotting the statistic error under different resolution or truncation by different lines under the same coordinate systems, which can give reader a more intuitive understanding of the relationship between error and K/N/training sample size
3. The paper deliberately avoids dealing with nonlinearity, focusing only on the linear core of the FNO. While this simplifies the analysis and provides foundational insight, it limits direct implications for practical, multi-layer neural operators used in real applications
4. The discretization error analysis is tailored to functions sampled on uniform grids. This excludes many real-world cases where data is irregular or unstructured, which restricts the estimator's utility in practical operator learning settings

strength:
1. The paper makes contribution by formally identifying and separating the three major sources of error in operator learning (statistical, truncation, and discretization) which provides conceptual clarity and guides future work on neural operator analysis.
2. The statistical error bound is derived using a refined Rademacher complexity approach, giving bounds that are independent of the truncation level.

---

> ### Author Response · Authors · 2025-07-12
> **Author response to Reviewer Fggu**
>
> We thank the reviewer for the thoughtful and constructive feedback. We address each of the main comments below.
>
> 1. We have added additional experimental results in Appendix G and summarized key takeway and implications of our experiments for further theoretical analysis.
>
> 2. We have added a discussion in Section 5 outlining how our results might extend beyond the linear setting. In particular, we note that extending our statistical analysis to nonlinear architectures would require a more refined Rademacher analysis, similar to the one in [1]. We also explain the key obstacle introduced by pointwise nonlinearities in the FNO architecture, namely, the fact that $\sigma(f)\neq \sum_{j}\sigma(\langle f, e_j\rangle) e_j$ in infinite-dimensional settings, which breaks a key step in existing proofs.
>
> 3. We now explicitly acknowledge the limitation of our uniform grid assumption in Section 5. The assumption of a uniform grid is used only to control the numerical integration error when approximating Fourier coefficients via the DFT. In principle, this could be replaced by any integration scheme with vanishing error—e.g., Gaussian quadrature for structured grids or Monte Carlo methods for unstructured grids. We briefly outline these alternatives and note this as an important direction for future work.
>
>
> [1] Noah Golowich, Alexander Rakhlin, and Ohad Shamir. Size-independent sample complexity of neural networks. In Conference On Learning Theory, pp. 297–299. PMLR, 2018.

---

### Decision · Action_Editor_PfoX · 2025-07-30

**Recommendation:** Accept as is

**Audience:**

Yes

**Audience Explanation:**

Operator learning is a fundemanlt ML problem and the authors study aspects of it.,

**Claims And Evidence:**

Yes

**Claims Explanation:**

The authors study learning-theoretic aspect of linear FNO, focusing on  sample size, Fourier truncation and discretization. The paper provides insight in theoretical and empirical designs of FNOs.